# Co-expression of a PD-L1-specific chimeric switch receptor augments the efficacy and persistence of CAR T cells via the CD70-CD27 axis

Le Qin[1], Yuanbin Cui[1], Tingjie Yuan[2,3], Dongmei Chen[1], Ruocong Zhao[4], Shanglin Li[1], Zhiwu Jiang[1], Qiting Wu[1], Youguo Long[1], Suna Wang[1], Zhaoyang Tang[5], Huixia Pan[5], Xiaoping Li[6], Wei Wei[7], Jie Yang[8], Xuequn Luo[9], Zhenfeng Zhang[10], Qiannan Tang[11], Pentao Liu[11], Robert Weinkove[12], Yao Yao[1], Dajiang Qin[2], Jean Paul Thiery[3] ✉ & Peng Li[1,4,13] ✉

Co-expression of chimeric switch receptors (CSRs) specific for PD-L1 improves the antitumor effects of chimeric antigen receptor (CAR) T cells. However, the effects of trans-recognition between CSRs and PD-L1 expressed by activated CAR T cells remain unclear. Here, we design a CSR specific for PD-L1 (CARP), containing the transmembrane and cytoplasmic signaling domains of CD28 but not the CD3 ζ chain. We show that CARP T cells enhance the antitumor activity of anti-mesothelin CAR (CARMz) T cells in vitro and in vivo. In addition, confocal microscopy indicates that PD-L1 molecules on CARMz T cells accumulate at cell-cell contacts with CARP T cells. Using single-cell RNA-sequencing analysis, we reveal that CARP T cells promote CARMz T cells differentiation into central memory-like T cells, upregulate genes related to Th1 cells, and downregulate Th2-associated cytokines through the CD70-CD27 axis. Moreover, these effects are not restricted to PD-L1, as CAR19 T cells expressing anti-CD19 CSR exhibit similar effects on anti-PSCA CAR T cells with truncated CD19 expression. These findings suggest that target trans-recognition by CSRs on CAR T cells may improve the efficacy and persistence of CAR T cells via the CD70-CD27 axis.

The programmed cell death receptor-1 (PD-1)/programmed cell death-ligand 1 (PD-L1) axis is considered one of the most important immunosuppressive signaling pathways for tumor evasion[1]. The PD-1 receptor is expressed at very low levels in resting T cells, but its expression can be induced following T cell activation and is also observed on activated B cells and myeloid cells[2]. Currently, two PD-1 ligands (PD-L1 and PD-L2) have been identified. PD-L1 upregulation is detected in multiple solid tumors. Previous studies have demonstrated that PD-L1 expression on cancer cells mediates an immunosuppressive

function through its interaction with PD-1 on T cells, ultimately resulting in T cell exhaustion, whereas PD-L2 expression is restricted mainly to dendritic cells (DCs) and macrophages[3–5]. To date, various types of immune checkpoint inhibitors targeting the PD-1/PD-L1 axis have been approved for the treatment of several tumors, and durable tumor control and acceptable safety profiles have been achieved[6,7].

CD70 is transiently upregulated on activated T cells, while its receptor, CD27, is physiologically expressed on T cells[8]. CD27/CD70 costimulation enhances T cell proliferation and survival and promotes

naïve T cell differentiation into antigen-specific cytotoxic and memory T cells[9–11]. The CD70-CD27 axis also promotes Th1 cell differentiation[12]. Th1 cells produce abundant proinflammatory cytokines, including IL2, IFN-γ, and TNF. Conversely, anti-inflammatory cytokines, such as IL5, IL10, and IL13, are secreted by Th2 cells[13,14]. Varlilumab, a CD27 agonistic antibody, shows promising efficacy in multiple cancer types[15].

Chimeric antigen receptor (CAR) T cell therapy has shown promising efficacy in the clinical management of B cell-derived malignancies, but not in solid tumors[16]. One of challenges of CAR T cell therapy against solid tumor is its poor persistence[17]. PD-L1 expression on solid tumors may be responsible for the poor efficacy of CAR T cells. To convert the inhibitory signaling initiated by PD-L1 into a stimulatory one, several groups co-expressed chimeric switch receptors (CSRs) targeting PD-L1 in CAR T cells and found that these CSRs augment the antitumor effects of CAR T cells[18,19]. Notably, PD-L1 is also expressed in activated T cells[20–22]. However, the effect of trans-recognition between CSRs and PD-L1 in activated CAR T cells on antitumor activity of CAR T cells remain to be investigated.

Here, to examine the effects of the trans-recognition between CSRs and PD-L1 on CAR T cells and the molecular mechanisms underlying these effects, we design a CSR targeting PD-L1 containing the transmembrane and cytoplasmic signaling domains of CD28, without the CD3ζ chain (CARP). T cells overexpressing CARP improve the antitumor activity of anti-mesothelin (MSLN) CAR (CARMz) T cells in vitro and in vivo and promote them differentiation into central memory-like T cells. In addition, these effects are not restricted to any specific antigens that the CSR recognize but depend on the ligation between CD27 on CARMz T cells and CD70 on CARP T cells. These findings suggest that the trans-recognition between CSR and target antigen on CAR T cells may enhance the antitumor activity and promote central memory-like CAR T cells formation via the CD70-CD27 axis.

## Results

### CARP T cells enhance the antitumor activity of CARMz T cells

A previous study demonstrated that coexpression of a CSR targeting PD-L1 (PD1CD28) augment the antitumor effects of CAR T cells[18,19]. As CAR T cells upregulate PD-L1 expression upon CAR activation[20,22], it remains unclear whether PD1CD28-expressing T cells can interact with antigen-stimulated CAR T cells and regulate the antitumor effects of the CAR T cells. To answer this question, we designed a CAR vector targeting PD-L1, referred to as CARP, consisting of a scFv against human PD-L1 (3208)[23], an intracellular domain that contained only the CD28 costimulatory domain without the CD3ζ chain, and a truncated CD19 (tCD19) tag, and a CAR vector targeting MSLN, named CARMz, containing a scFv against human MSLN (SS1)[24], the CD28 costimulatory domain, the CD3ζ chain, and a GFP tag. CAR19z, which contains a FMC63 scFv[25], the CD28 costimulatory domain and the CD3ζ chain, served as a negative control (Supplementary Fig. 1a). We transduced these CAR vectors individually into T cells (Supplementary Fig. 1b). CARP T cells were cocultured with K562-PDL1-GL cells, a leukemia cell line that was negative for MHC-I molecules and engineered to express PDL1-GL, a vector containing PD-L1, GFP and luciferase (Supplementary Fig. 2a, b)[26]. We found that the percentage of CD25⁺CD69⁺ cells in CARP T cells and the amount of IL2 and IFN-γ produced by CARP T cells were increased in the co-culture, compared with those of CAR19z T cells (Supplementary Fig. 2c, d). However, CARP T cells did not lyse K562-PDL1-GL cells as it is also observed with the control CAR19z cells (Supplementary Fig. 2e). These results show that CARP T cells could be activated by tumor target cells expressing PD-L1 but did not lyse them in vitro. In contrast, CARMz T cells efficiently lysed HeLa-GL cells, a cervical cancer cell line that spontaneously expresses MSLN but not PD-L1, and was engineered to express GFP and luciferase (GL) (Supplementary Figs. 2a, 3a–c). In line with previous studies[20], CARMz T cells exhibited upregulated PD-L1 at 16 h post-coculture with

HeLa-GL cells but its expression attenuated at 48 h post-coculture (Supplementary Fig. 3d, e).

To study the interaction between CARP T and CARMz T cells, we then mixed them at a ratio of 1:1 and cocultured them with HeLa-GL cells. We found that the mixture of CARMz T and CARP T cells lysed HeLa-GL cells more efficiently than CARMz T cells alone, while CARP T cells alone were unable to lyse these target cells (Fig. 1a). In addition, the mixed CAR T cells secreted significantly more antitumor cytokines, such as IL-2 and IFN-γ, than CARMz T cells alone, while CARP T cells alone had only a modest production IFN-γ (Fig. 1b). Similarly, the mixture of CARMz T and CARP T cells lysed H460-MSLN-GL cells, a lung cancer cell line that highly expresses both MSLN and PD-L1 (Supplementary Figs. 2a, 4a), more efficiently and secreted more IL-2 and IFN-γ than separated cultures of CARMz T cells, while CARP T cells only had mild killing activity to target cells and produced very minimal amounts of cytokines (Supplementary Fig. 4b, c). Interestingly, CARP T cells enhanced the killing capacity of CARMz T cells not through secreting IL-2 or IFN-γ, as the mixed CAR T cells still exhibited augmented cytotoxicity in the presence of anti-IL2 or anti-IFN-γ monoclonal antibodies (Supplementary Fig. 4d).

We next compared the antitumor effects of a combination of CARMz T and CARP T to CARMz T cells alone in vivo. In mice bearing HeLa-GL xenografts, both the tumor volumes and weights of the combination group were lower than those of the CARMz T cell group (Fig. 1c, d). Of interest, we found that CARMz T cells in the tumors from the combination group exhibited phenotypes of central memory T cells (Tcm, CD45RO⁺, CCR7⁺) and stem cell memory T cells (Tscm, CD45RO⁻, CCR7⁺), while CARMz T cells in the tumors from the separated CARMz T cell group were mainly effector T cells (CD45RO⁻, CCR7⁻) and effector memory T cells (CD45RO⁺, CCR7⁻) (Fig. 1e, f and Supplementary Fig. 9a). Similar results were obtained when we increased the doses of CAR T cells from $2.5 \times 10^6$ per mouse to $5 \times 10^6$ per mouse (Supplementary Fig. 5a, b). We also assessed the antitumor activity of the combination of CARMz T and CARP T cells in a PD-L1⁺MSLN⁺ NSCLC PDX mouse model (Supplementary Fig. 5c). In line with the results in HeLa-GL xenografts, the tumors in the combination group were significantly smaller and lighter than those in the CARMz T cell and CARP T cell groups (Supplementary Fig. 5d, e). Altogether, these results show that the combination of CARP T and CARMz T cells was more efficient in repressing tumor growth than CARMz T cells alone. In addition, CARP T cells might promote memory T cell formation in CARMz T cells, as tumor-infiltrated CARMz T cells from the combination group contained central memory-like T cells.

### Formation of cell-cell contacts between CARMz T and CARP T cells

To test whether CARP T cells promote the antitumor effects of CARMz T cells simply through blockade of PD-L1 on CARMz T cells, similar to anti-PD-L1 antibodies, such as atezolizumab (AZ)[27], we cocultured CARMz T cells with HeLa-GL cells with or without AZ and found that AZ did not enhance the killing capacity of CARMz T cells (Fig. 2a). Similarly, anti-PD-1 mAb, pembrolizumab, did not elevate the lysing capacity of CARMz T cells against HeLa-GL cells (Fig. 2b). These results suggest that blockage of PD-L1 or PD-1 did not improve cytotoxicity of CARMz T cells in vitro. Interestingly, treatment of CARMz T cells with AZ did reduce the tumor-lysing capacity of the mixture of CARMz T and CARP T cells (Fig. 2c), suggesting that CARP T cells augmented cytotoxicity of CARMz T cells possibly through physical interactions between PD-L1 on CARMz T cells and CSR on CARP T cells (Fig. 2d).

To confirm this hypothesis, we co-cultured CARMz T cells that had been transiently activated by HeLa cells before being co-cultured with CARP T cells and observed the formation of contacts between these two types of T cells by confocal microscopy (Fig. 2e). In particular, PD-L1 molecules on CARMz T cells that expressed GFP concentrated at the junctions between CARMz T and CARP T cells (Fig. 2e). Conversely,

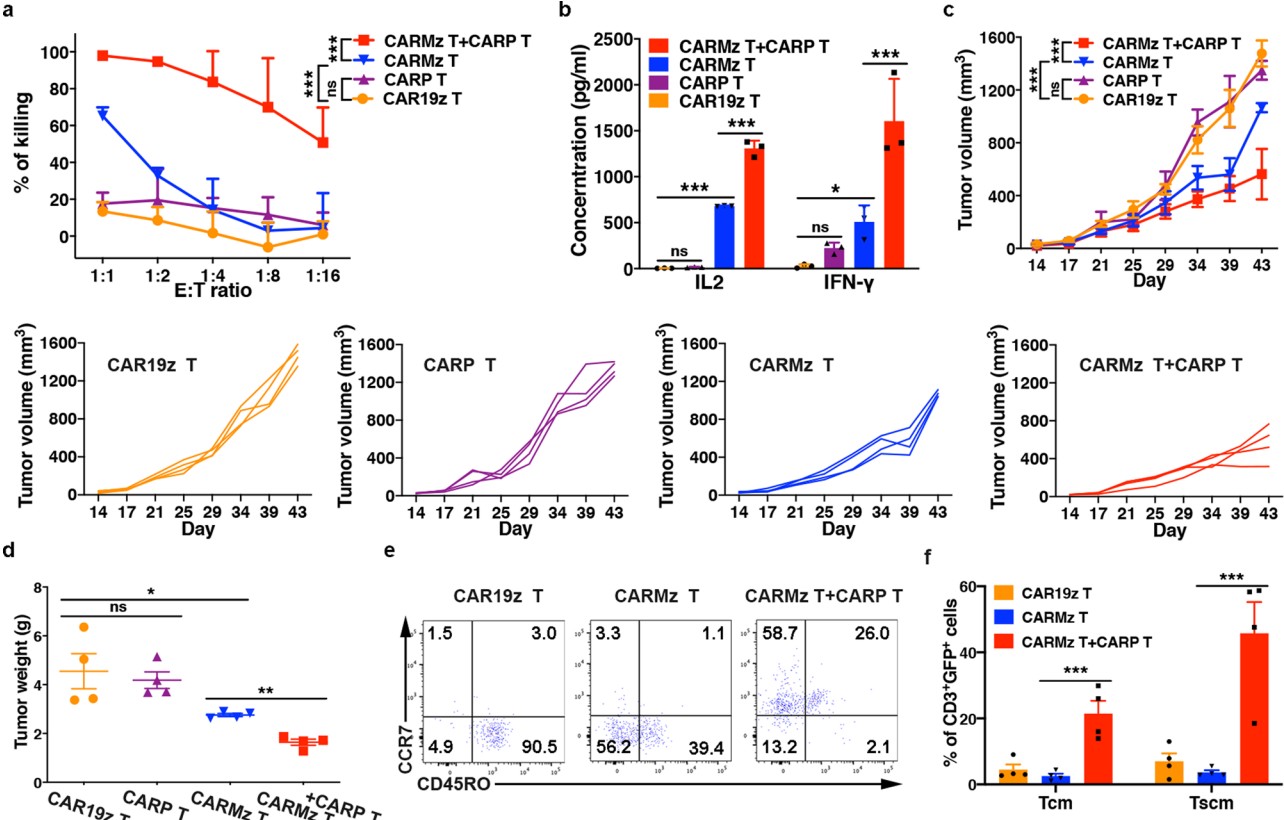

**Fig. 1 | CARP T cells enhance the antitumor efficacy of CARMz T cells. a** CARP T, CARMz T, a mixture of CARMz T and CARP T and control CAR19z T cells cytotoxicity against HeLa-GL cells were measured at various E:T ratios. Data are presented as mean ± SD ($N = 3$ independent experiments). *p* Values (CAR19z T vs. CARP T = 0.115, CAR19z T vs. CARMz T = 2.903E−04, CARMz T + CARP T vs. CARMz T = 4.92E−13). **b** The production of IL-2 and IFN-γ by CARP T, CARMz T, a mixture of CARMz T and CARP T and control CAR19z T cells. Data are presented as mean ± SD ($N = 3$ biological samples). *p* Values (CAR19z T vs. CARP T = IL2: 0.757, IFN-γ: 0.360, CAR19z T vs. CARMz T = IL2: 5.795E−08, IFN-γ: 0.044, CARMz T + CARP T vs. CARMz T = IL2: 1.035E−07, IFN-γ: 6.386E−04). **c**−**f** NSI mice bearing HeLa-GL tumors ($5 \times 10^5$, established for 14 days) were infused with CARP T, CARMz T, a mixture of CARMz T and CARP T or CAR19z T cells ($2.5 \times 10^6$). **c** Tumor volumes were monitored on the indicated days. Individual tumor responses to CAR-T cell injection are shown with spider plots below. Data are presented as mean ± SD ($N = 4$ mice per group). *p* Values (CAR19z T vs. CARP T = 0.999, CAR19z T vs. CARMz T = 4.282E−10, CARMz T + CARP T vs. CARMz T = 1.915E−06). **d** Tumor weights were measured after mouse euthanasia. *p* Values (CAR19z T vs. CARP T = 0.897, CAR19z T vs. CARMz T = 0.025, CARMz T + CARP T vs. CARMz T = 0.004). **e** Representative phenotype of CARMz T, CARMz T from a mixture of CARMz T and CARP T and control CAR19z T cells (gated on CD3+GFP+ cells) within tumors. **f** Proportions of Tcm (central memory T, CD45RO+CCR7+) and Tscm (stem cell memory T, CD45RO−CCR7+) cells in (**e**). *p* Values (CARMz T + CARP T vs. CARMz T = Tcm: 3.799E−04, Tscm: 5.043E−04). Data of **d** and **f** are presented as mean ± SEM ($N = 4$ mice per group). *p* Values of **a** and **c** were calculated by two-way ANOVA with Tukey's multiple comparisons test. *p* Values of **b**, **d**, and **f** were calculated by one-way ANOVA with Sidak's post hoc test. *$p < 0.05$, **$p < 0.01$, and ***$p < 0.001$.

---

CARMz T cells or CARP T cells did not form any connections between themselves in either co-culture or separated culture. In separate culture, PD-L1 molecules were evenly distributed on the surface of CARMz T cells (Fig. 2e). Altogether, these results suggest that CARP T cells augmented the cytotoxicity of CARMz T cells through physical interactions between PD-L1 on CARMz T cells and CSR on CARP T cells.

## CARP T cells promote CARMz T cell to differentiate to central memory-like T cells

To uncover the effects of physical interactions between CARMz T and CARP T cells on CARMz T cells, we purified CARMz T cells from a mixture of CARMz T and CARP T cells (mCARMz T) by flow cytometry based on GFP expression and compared their transcription profiles to those of CARMz T cells treated with AZ (CARMz T + AZ) and CARMz T cells (sCARMz T) that were cultured separately without CARP T cells by bulk RNA-seq analysis (Fig. 3a). There were 374 upregulated differentially expressed genes (DEGs) and 276 downregulated DEGs in mCARMz T cells compared to sCARMz T cells (Fig. 3b), suggesting that the transcriptomic profiles of CARMz T cells changed dramatically after coculture with CARP T cells. Conversely, there were only 1 upregulated DEGs and 11 downregulated DEGs in CARMz T cells upon AZ treatment (Fig. 3b). Further analysis with a heatmap shows that

mCARMz T cells specifically upregulated genes related to stem cell maintenance (*RIF1*, *IGF1*, and *KLF4*)[28–30], long-lasting T cell maintenance (*IL6ST*)[31], WNT pathway (*WNT10A*, *WNT10B*, *WNT16*, *GIPR*, *MST1*, *IRS2*, and *TCF7L2*) and antiapoptotic processes (*BCL6*, *NRN1*, and *NOL3*)[32–37] compared with sCARMz T cells and AZ-treated CARMz T cells (Fig. 3c). Of interest, the expression of *FGFBP2*, a Th1-specific gene[38], was also increased in mCARMz T cells (Fig. 3c). In addition, the expression of Th2-associated genes, including *IL5* and *IL13*, was significantly decreased in mCARMz T cells compared with sCARMz T cells and AZ-treated CARMz T cells (Fig. 3c)[39]. Gene set enrichment analysis (GSEA) also illustrates the enrichment of genes in WNT signaling pathway that plays a role in memory T cell formation in mCARMz T cells, compared to sCARMz T (Fig. 3d). In line with bulk RNA-seq results, the concentrations of IL5, IL10 and IL13 significantly decreased in the supernatant of cocultures of CARMz T and CARP T cells compared with those of cultures of CARMz T cells alone after activation with HeLa-GL cells (Fig. 3e). Furthermore, the level of intracellular IL13 expression in mCARMz T cells was also lower than that in sCARMz T cells or AZ-treated CARMz T cells (Fig. 3f and Supplementary Fig. 9b). Taken together, these results suggest that CARP T cells facilitated upregulation of genes that are related to memory T cell and Th1 cell and downregulation of Th2 cell-related genes in CARMz T cells.

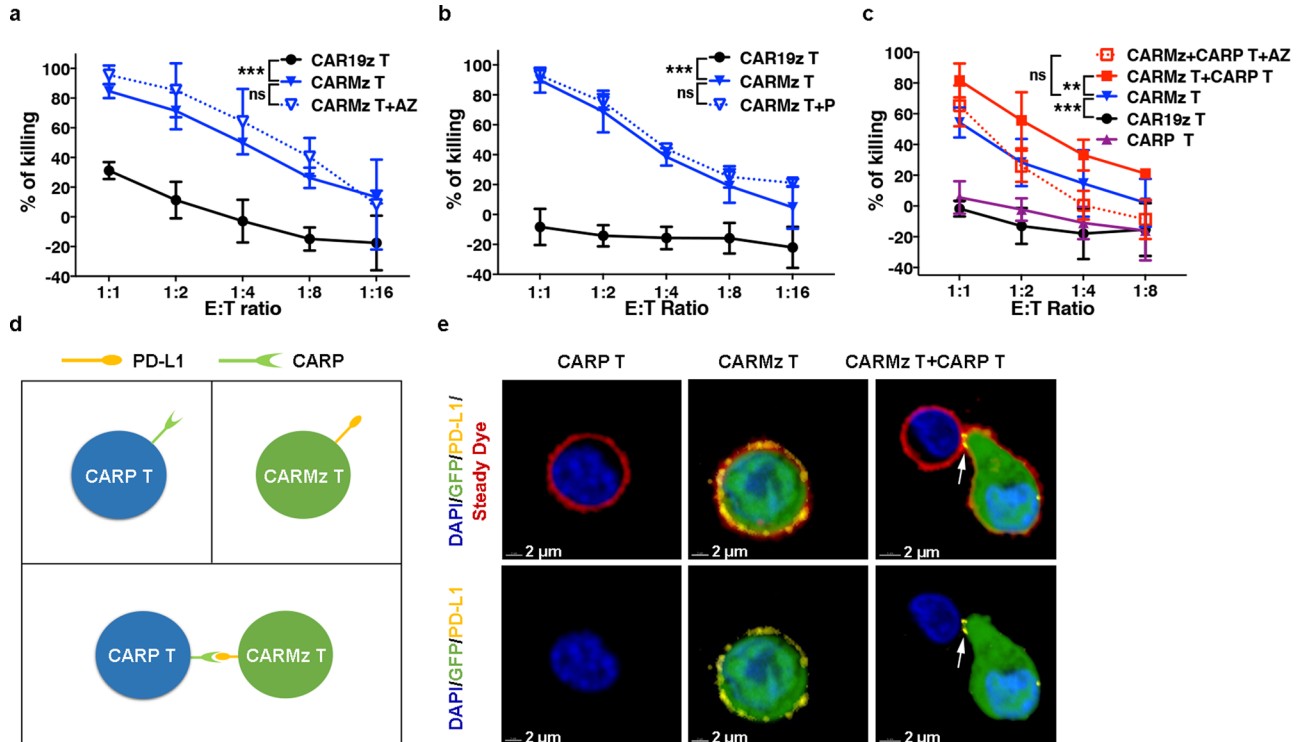

**Fig. 2 | Formation of cell–cell connections between CARMz T and CARP T cells.** **a** CARMz T and CAR19z T cells with or without an anti-PD-L1 mAb (AZ, 20 µg/ml) against HeLa-GL cells were measured at various E:T ratios. *p* Values (CAR19z T vs. CARMz T = 7.499E−10, CARMz T + AZ vs. CARMz T = 0.164). **b** CARMz T and CAR19z T cells with or without an anti-PD-1 mAb (P, 20 µg/ml) against HeLa-GL cells were measured at various E:T ratios. *p* Values (CAR19z T vs. CARMz T = 5E−14, CARMz T + P vs. CARMz T = 0.084). **c** CARMz T, CARP T, a mixture of CARMz T and CARP T, a mixture of CARMz T and CARP T treated with anti-PD-L1 mAb (AZ, 20 µg/ml) and control CAR19z T cells against HeLa-GL cells were measured at various E:T ratios. *p* Values (CAR19z T vs. CARMz T = 3.309E−07, CARMz T + CARP T vs. CARMz T = 0.001, CARMz T + CARP T + AZ vs. CARMz T = 0.939). Data of **a**–**c** are presented as mean ± SD (*N* = 3 independent experiments). *p* Values of **a**–**c** were calculated by two-way ANOVA with Tukey's multiple comparisons test. **d** Schematic diagram of the interaction between CARMz T and CARP T cells. Individual CARP T cells (upper left), individual CARMz T cells with PD-L1 expression (upper right), and the mixture of CARMz T and CARP T cells (below). **e** CARMz T, CARP T and a mixture of CARMz T and CARP T cells were stained for nuclei (blue), cell membrane (red), and PD-L1 (yellow) after incubate with HeLa cells (*N* = 4 independent experiments and three representative pictures were presented). GFP staining corresponds to CARMz T cells. The white arrow indicates cell-cell contacts between CARP and PD-L1 molecules. ***p < 0.001.

To further characterize mCARMz and sCARMz T cells at single cell levels, we performed single cell RNA-seq (scRNA-seq) analysis and collected 4264 individual CARMz T cells, including 1989 sCARMz T cells and 2275 mCARMz T cells, and detected a mean of 1964 gene transcripts in sCARMz T cells and 1667 gene transcripts in mCARMz T cells (Fig. 4a). PCA and uniform manifold approximation and projection (UMAP) dimension reduction were performed to analyze these scRNA-seq data. The Louvian modularity optimization algorithm was then applied to iteratively classify cells together into eight clusters that were visualized in UMAP (clusters 1–8) (Fig. 4b). sCARMz and mCARMz T cells displayed distinct distributions across the eight clusters (Fig. 4b, c). T-distributed stochastic neighbor embedding (tSNE) dimension reduction analysis scRNA-seq data also yielded similar cell clustering (Supplementary Fig. 6). Cells from the cluster 1 (C1) did not express genes related to T cell activation, proliferation, cytotoxicity, or memory formation, suggesting that they were non-activated T cells (Tua) (Fig. 4d, e). Cells from C2 were CD8[+] effector cells (Tef8), as they expressed genes related to effector T cell functions (*CCL5, GZMA, PRF1, GNLY,* and *IFNG*)[40–42] and the AP1 family (*JUN*) (Fig. 4d, e)[43]. Cells from C6 highly expressed T cells activation and costimulation associated genes (*CD28* and *TNFRSF4*) and *JUN*, thus were considered as Tef4 (Fig. 4d, e)[43–45]. Since the cells in C3 and C7 exhibited moderate expression of genes associated with naive T cells (*LEF1* and *CD27*) and effector T cells (*GZMA*) (Fig. 4d, e)[46–48], they might represent T cells at an intermediate transition state between the naive and effector states. We defined C3 and C7 as CD8[+] and CD4[+] partially differentiated effector T cells (Tie8 and Tie4), respectively.

Cells from C4, C5, and C8 highly expressed *CCR7, CD27,* and *LEF1*[49–51]. In addition, cells from C5 also expressed higher levels of *CD28* and *JUN*, compared to cells from C4 (Fig. 4d, e)[43,44]. Therefore, we named C4 and C8 as CD8[+] and CD4[+] central memory-like T cells (Tcm8 and Tcm4)[52–54], respectively, and C5 as CD8[+] activated central memory-like T cells (acTcm8).

GSEA analysis also indicated that genes that are highly expressed in CD8[+] stem cell memory T cells but not in CD8[+] naive T cells were enriched in cells from the cluster Tcm8 (Fig. 4f). In addition, genes that are upregulated in CD4[+] effector memory T cells, compared to CD4[+] central memory T cells were negatively enriched in cells from the cluster Tcm4 (Fig. 4g). Most importantly, the numbers of mCARMz T cells were higher than those of sCARMz T cells in Tcm4 (235 mCARMz T cells vs. 83 sCARMz T cells) and Tcm8 (478 mCARMz T cells vs. 287 sCARMz T cells) (Fig. 4h), suggesting that mCARMz T cells contained more central memory-like T cells than sCARMz T cells. Consistent to scRNA-seq analysis (Fig. 4h), the percentages of CD8[+] and CD4[+] central memory T cells in mCARMz T cells were higher than those in sCARMz T cells, respectively (Fig. 4i, j and Supplementary Fig. 9c). Taken together, the scRNA-seq results indicate that CARP T cells promoted CARMz T cells to differentiate to central memory-like T cells and Th1 cells.

**CARMz T cells activate CARP T cells into effector T cells**
To study the effects of CARMz T cells on CARP T cells, we sorted CARP T cells (mCARP T cells) from a mixture of CARMz T and CARP T cells by

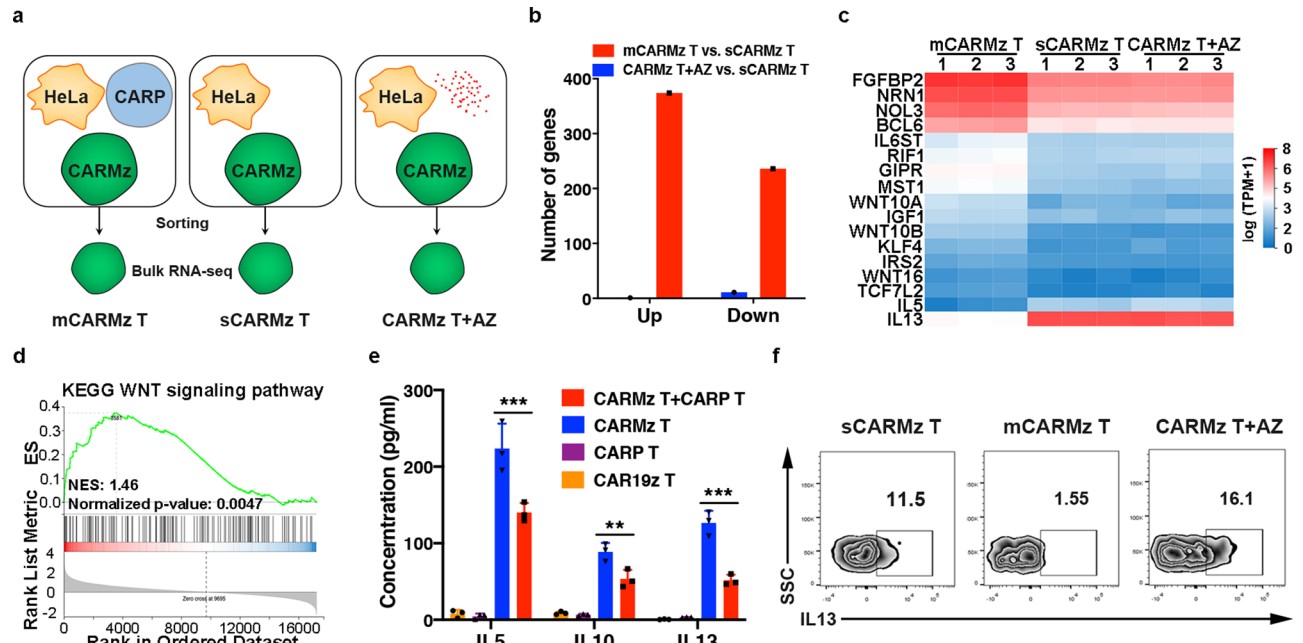

**Fig. 3 | Bulk RNA-seq analysis of individual CARMz T (sCARMz T), CARMz T from a mixture of CARMz T and CARP T (mCARMz T) and CARMz T cells treated with anti-PD-L1 mAb (CARMz T + AZ) after coculture with HeLa-GL cells. a–d** Bulk RNA-seq strategy: individual CARMz T (sCARMz T), CARMz T from a mixture of CARMz T and CARP T (mCARMz T) and CARMz T cells treated with AZ (20 µg/ml) (CARMz T + AZ) were isolated by flow cytometry sorting based on their GFP tag post-coculture with HeLa-GL cells, and then performed bulk RNA-seq analysis ($N = 3$ biological samples). The red dots represent AZ. **b** The numbers of genes upregulated and downregulated in CARMz T + AZ and mCARMz T compared with sCARMz T. **c** The heatmap shows clustering of differential expressed genes (DEGs) in sCARMz T, mCARMz T, and CARMz T + AZ ($N = 3$ biological samples). Cutoff: absolute log2 (fold change) ≥ 1; adjusted $P$ value ≤0.05. **d** GSEA illustrating the enrichment of genes in WNT signaling pathway in mCARMz T versus sCARMz T. NES (Normalized Enrichment Scores) and normalized $p$-values are indicated. **e** The production of IL5, IL10, and IL13 by CARP T, CARMz T, a combination of CARMz T and CARP T and control CAR19z T cells post-coculture with HeLa-GL cells. Data are presented as mean ± SD ($N = 3$ independent experiments). $p$ Values were calculated by two-way ANOVA with Tukey's multiple comparisons test (CARMz T + CARP T vs. CARMz T = IL5: 1.016E−07, IL10: 0.009, IL13: 7.235E−07). **f** Percentage of IL13-secreting cells in individual CARMz T (sCARMz T), CARMz T from a mixture of CARMz T and CARP T (mCARMz T) and CARMz T cells treated with AZ (20 µg/ml) (CARMz T + AZ) post-coculture with HeLa-GL cells at a 1:1 E:T ratio for 24 h (gated on CD3⁺GFP⁺ cells). *$p < 0.05$, **$p < 0.01$ and ***$p < 0.001$.

flow cytometry based on their tCD19 expression and compared their transcription profiles to separated CARP T cells (sCARP T) using bulk RNA-seq analysis (Supplementary Fig. 7a). The RNA-seq analysis shows that mCARP T cells slightly upregulated genes related to T cell activation (*IL2RA* and *CD69*)[55,56], cytotoxicity (*TNF*, *GZMB*, and *IL2*) migration (*CXCL8*, *CCL3*, and *CCL4*)[57] and genes in the AP-1 family (*FOS* and *JUN*) compared with sCARP T cells (Supplementary Fig. 7b). In line with the RNA-seq results, mCARP T cells exhibited a higher percentage of surface markers associated with T cell activation, such as 4-1BB and CD28, than sCARP T cells (Supplementary Fig. 7c). Thus, these results demonstrated that the activation status of CARP T cells was enhanced once they encountered activated CARMz T cells.

We also used scRNA-seq to determine the single-cell transcriptional profiles of mCARP T cells and sCARP T cells post-coculture with HeLa-GL cells (Supplementary Fig. 7a). scRNA-seq results were obtained for a total of 3924 individual single cells: 1196 sCARP T cells and 2728 mCARP T cells. We detected a median of 1369 genes and a minimum of at least 229 genes in each cell. Details on scRNA-seq data processing are provided in the methods section. A total of seven distinct clusters (C1−7) were identified in UMAP diagram (Supplementary Fig. 7d), with completely different distribution characteristics for sCARP and mCARP T cells (Supplementary Fig. 7e). We also analyzed scRNA-seq data abased on tSNE dimension reduction method and obtained similar cell clustering (Supplementary Fig. 7f, g). Notably, clusters C1, C4, and C7 were mainly composed of sCARP T cells (Supplementary Fig. 7h). In particular, 90% of the cells in C4, 89% of the cells in C1 and 92% of the cells in C7 were sCARP T cells (Supplementary Fig. 7h). In contrast, the majorities of the other clusters (97% of the cells in C5, 98% of the cells in C2, 91% of the cells in C6, 98% of the cells in C3) were mCARP T cells

(Supplementary Fig. 7h). Moreover, the cells from clusters C4−C7 were mainly CD4⁺ T cells, and the cells from C1-C3 were mainly CD8⁺ (Supplementary Fig. 7i). The T cells from clusters C4 and C1 highly expressed *TCF7* but expressed *IL2RA*, *TNFRSF4*, and *CD70* at low levels[8], suggesting that they were non-activated CD4⁺ and CD8⁺ T cells (Tua4 and Tua8), respectively (Supplementary Fig. 7i, j). The T cells from clusters C5 and C2 were considered as CD4⁺ and CD8⁺ effector T cells (Tef4 and Tef8), respectively, as they highly expressed genes related to T cell activation (*IL2RA*, *CD70*, *TNFRSF4*, and *JUN*) (Supplementary Fig. 7i, j). In addition, the T cells from C2 highly expressed effector T cell-related genes (*GZMA*, *GZMB*, *IFNG*, and *CCL5*) (Supplementary Fig. 7i, j)[58]. T cells in cluster C7 were classified as CD4⁺ Th17 T cells (T17), as they highly upregulated the expression of *CASP1* and *IL26*, which are key markers of Th17 T cells (Supplementary Fig. 7i, j)[59,60]. Since the cells in C6 and C3 exhibited moderate expression of genes associated with naive T cells (*TCF7*) and effector T cells (*FOS*, *JUN*, *GZMA*, and *GZMB*)[61], they might represent T cells at an intermediate transition state from the naive to effector state (Supplementary Fig. 7i, j). We thus defined C6 and C3 as partially differentiated CD4⁺ and CD8⁺ effector T cells (Tie4 and Tie8), respectively. Taken together, these scRNA-seq analysis results show that mCARP T cells contained higher percentages of effector T cells and partially differentiated effector T cells than sCARP T cells, suggesting that CARMz T cells augmented the activation of CARP T cells.

## The CD70-CD27 axis is indispensable for the promotion of cytotoxicity of CARMz T cells by CARP T cells

To dissect how CARP T cells promote CARMz T cells to differentiate to memory T cells, we quantitatively analyzed intercellular communication networks based on scRNA-seq data of CARMz T and CARP T cells

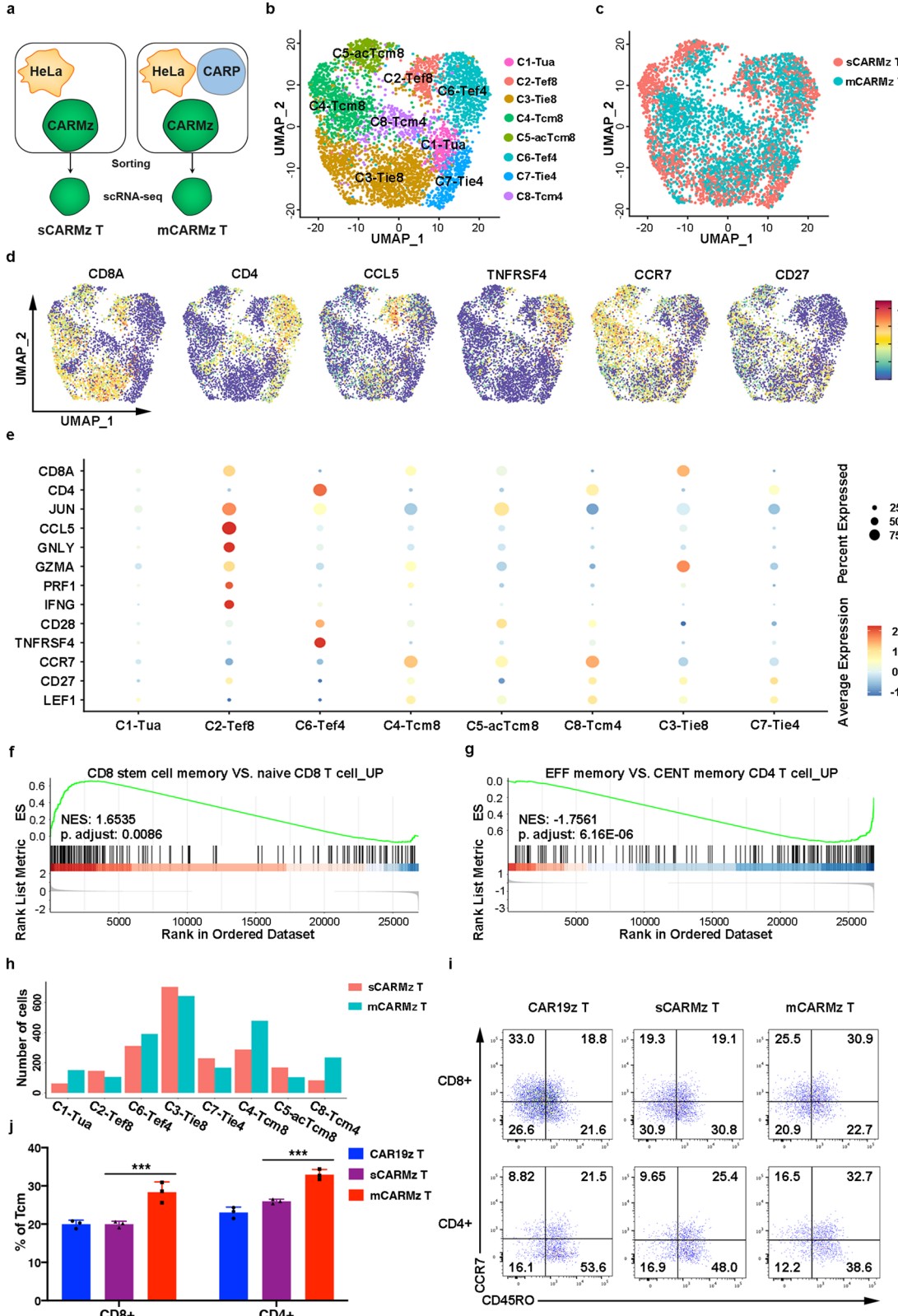

by CellChat tool[62]. As there were more CD4[+] and CD8[+] central memory-like CARMz T cells (M-Tcm4 and M-Tcm8, respectively) in co-culture with CARP T cells, compared to the culture without CARP T cells (Fig. 4h), we first analyzed potential interactions between the M-Tcm4 cluster with other subsets of CARMz T and CARP T cells, and found that the interaction strengths of the M-Tcm4 cluster with CD8[+] effector CARP T cells (the P-Tef8 cluster) and CD8[+] non-activated CARP T cells (the P-Tua8 cluster) were stronger than those with other clusters

(Fig. 5a and Supplementary Table 1)[62]. Similarly, the M-Tcm8 cluster interacted strongly with these two clusters from CARP T cells (P-Tef8 and P-Tua8) (Fig. 5b and Supplementary Table 1). To uncover how the P-Tef8 and P-Tua8 clusters affect the M-Tcm4 and M-Tcm8 clusters, we analyzed the outgoing signaling pathways of P-Tef8 and P-Tua8 and the incoming signaling pathways of M-Tcm4 and M-Tcm8. The outgoing patterns revealed how the cells as signal sources coordinated with each other, as well as how they coordinated with certain signaling

**Fig. 4 | scRNA-seq analysis of individual CARMz T (sCARMz T) and CARMz T from a mixture of CARMz T and CARP T cells (mCARMz T) after coculture with HeLa-GL cells. a–h** Single cell RNA-seq strategy: Individual CARMz T (sCARMz T) and CARMz T from a combination of CARMz T and CARP T cells (mCARMz T) were separated by flow cytometry sorting based on their GFP tag post-coculture with HeLa-GL cells, and then processed for scRNA-seq. **b** The UMAP projection of single CARMz T from sCARMz T and mCARMz T cells. C1-Tua: non-activated T cells, C2-Tef8: CD8$^+$ effector T cells, C3-Tie8: partially differentiated CD8$^+$ effector T cells, C4-Tcm8: CD8$^+$ central memory T cells, C5-acTcm8: activated CD8$^+$ central memory T cells, C6-Tef4: CD4$^+$ effector T cells, C7-Tie4: partially differentiated CD4$^+$ effector T cells, C8-Tcm4: CD4$^+$ central memory T cells. Each dot corresponds to a single cell. **c** Single cells from sCARMz T (red) and mCARMz T (blue) clusters in distinct regions of the UMAP space. **d** Single-cell transcript levels of CD8, CD4, CCL5, TNFRSF4, CCR7, and CD27 illustrated in UMAP plots. **e** Dot plot of selected DEGs expressed in each cluster. **f, g** GSEA illustrating the enrichment of genes defined as "CD8 stem cell memory vs. naive CD8 T cell UP" in cluster Tcm8 and genes defined as "effector memory vs. central memory CD4 T cell UP" in cluster Tcm4. NES (Normalized Enrichment Scores) and $p$ adjust are indicated. **h** Bar graphs summarizing the number of sCARMz T and mCARMz T cells in each cluster. **i** Representative phenotype of separated CARMz T (sCARMz T), CARMz T from a mixture of CARMz T and CARP T (mCARMz T), and control CAR19z T cells (gated on CD8$^+$GFP$^+$ cells and CD8$^-$GFP$^+$ cells) post-coculture with HeLa cells for 36 h. **j** Proportion of Tcm (central memory T cells, CD45RO$^+$CCR7$^+$) in sCARMz T, mCARMz T and control CAR19z T cells in (**i**). Data are presented as mean ± SD ($N = 3$ biological samples). $p$ Values were calculated by two-way ANOVA with Tukey's multiple comparisons test (mCARMz T vs. sCARMz T = CD8+: 3.913E−05, CD4+: 2.268E−04). ***$p < 0.001$.

pathways to drive communication. Incoming patterns showed how the cells as signal receivers coordinated with each other, as well as how they coordinated with certain signaling pathways to respond to incoming signals[62]. In total, 18 outgoing or incoming signaling pathways were identified, including four incoming signaling pathways that were expressed in M-Tcm4 and M-Tcm8 clusters simultaneously (LT, CD70, ALCAM, and JAM) (Fig. 5c). Within these four incoming signaling pathways, CD27, the corresponding ligand of CD70, was expressed in M-Tcm4 and M-Tcm8 clusters, while only CD70 was expressed in the P-Tef8 cluster (Fig. 5c, d). T cells in the P-Tua8 cluster did not express any corresponding outgoing signaling pathways of LT, CD70, ALCAM, or JAM pathways (Fig. 5c, d). These results indicated that CARP T cells from the P-Tef8 clusters promoted CARMz T cells to differentiate to memory-like T cells through the CD70-CD27 axis.

Previous studies show that the CD70-CD27 axis promotes the formation and maintenance of memory T cells and is essential for Th1 cell differentiation[8,12]. CD27 was highly expressed on both CD4$^+$ and CD8$^+$ CARMz T cells (Fig. 6a, b and Supplementary Fig. 10a), while CD8$^+$ CARP T cells also expressed CD70 (Fig. 6c, d, Supplementary Fig. 10b). To validate whether CARP T cells promote CARMz T cells to differentiate into memory-like T cells through the CD70-CD27 axis, we blocked this axis with anti-CD70 mAb (αCD70) and found that αCD70 treatment reduced the percentages of central memory T cells defined as CD45RO$^+$CCR7$^+$ in mCARMz T cells (Fig. 6e, f and Supplementary Fig. 9c). In addition, the tumor lysing capacity of CARMz T cells in the mixture of CARMz T and CARP T cells was compromised upon αCD70 treatment in vitro (Fig. 6g), suggesting that CARP T cells augmented the cytotoxicity of CARMz T cells through the interaction between CD70 and CD27. Taken together, these results show that the CD70-CD27 axis is essential for CARP T cells to promote CARMz T cells to differentiate into central memory-like CARMz T cells and to lyse tumor targets.

### CARP T cells improve the antitumor effects of CAR19z T cells

To test whether CARP T cells can also promote CAR T cells targeting other antigens besides MSLN to differentiate into central memory-like T cells (Supplementary Fig. 8a), we cocultured separated CAR19z T cells (sCAR19z T) or a mixture of CAR19z T and CARP T cells (mCAR19z T) with CD19$^+$ NALM6-GL cells (Supplementary Fig. 8b). We found that the percentages of central memory T cells in CD8$^+$ and CD4$^+$ mCAR19z T cells were higher than those in CD8$^+$ and CD4$^+$ sCAR19z T cells, respectively (Supplementary Fig. 8c, d). These results demonstrate that the effects of CARP T cells on CAR T cells are not restricted to their targeting antigens.

### CAR19 T cells augment the efficacy of CARPAz T cells

As the effects of CARP T cells on CAR T cells depends on the binding between CSR on CARP T cells and PD-L1 molecules that were transiently expressed upon activation on CARMz T cells, we wondered whether any binding between a CSR and its antigen can facilitate

similar effects of CARP T cells on CAR T cells. We thus modified the CARP vector by replacing the scFv of 3208 with a scFv of FMC63 targeting CD19 instead of PD-L1. Similar to the CARP vector, the new CSR vector, named as CAR19, contained a CD28 costimulatory domain but not CD3ζ chain (Supplementary Fig. 1a, c). We also designed a CAR vector against prostate stem cell antigen (PSCA), named CARPAz, containing the 1G8 scFv, the CD28 costimulatory domain, the CD3ζ chain, and a tCD19 tag that contained extracellular and transmembrane domain of natural CD19 (Supplementary Fig. 1a, c). The CSR on CAR19 T cells could bind the tCD19 on CARPAz T cells (Fig. 7a).

Similar to the co-culture of CARMz T and CARP T cells, the mixture of CARPAz T (mCARPAz T) and CAR19 T cells lysed HeLa-GL cells, that highly expressed PSCA (Supplementary Fig. 8e), more efficiently than CARPAz T cells (sCARPAz T) and CAR19 T cells evaluated separately (Fig. 7b). We also found that mCARPAz T cells secreted less amounts of IL13 than sCARPAz T cells, suggesting that CARP T cells inhibited IL13 secretion in CARPAz T cells (Fig. 7c and Supplementary Fig. 10c), which is in line with the effects of CARP T cells on CARMz T cells (Fig. 3f).

To assess whether CAR19 T cells can also improve the antitumor activity of CARPAz T cells in vivo, we infused CARPAz T cells and CAR19 T cells separately or in combination into HeLa-GL xenografts. Compared to CAR19 T or CAR19z T cells as negative controls, CARPAz T cells and the mixture of CARPAz T and CAR19 T cells inhibited tumor growth (Fig. 7d, e). In line with previous results (Fig. 1c, d), the tumor volumes and weights of the combination group were significantly lower than those of the CARPAz T cell group (Fig. 7d, e). Taken together, these results demonstrate that CAR19 T cells augment antitumor effects of CARPAz T cells, indicating that connection between CSR molecules and their antigens can indirectly improve antitumor activity of CAR T cells.

### Discussion

In this study, we found that anti-PD-L1 CSR (CARP) T cells not only increased the percentages of central memory-like T cells of anti-mesothelin CAR (CARMz) T cells but also improve antitumor activity of CARMz T cells in vitro and in vivo. In addition, similar phenotypes were also observed when we replaced the anti-PD-L1 CSR with an anti-CD19 CSR that recognized tCD19 on anti-PSCA CAR (CARPAz) T cells. These results suggest that the effects of CSR-T cells on CAR T cells depend on the binding between CSR molecules on CSR-T cells and their antigens on CAR T cells but are not restricted to any specific antigens. Indeed, we observed a direct intercellular contact between CARP T and CARMz T cells. Of note, PD-L1 was polarized in the cell-cell connection, suggesting that this connection is initiated by the recognition between CSR and antigens on CAR T cells. What is the function of the cell-cell contact between CARP T and CARMz T cells? Are there any signaling communications between CSR-T and CAR T cells through the cell-cell connection? Further investigations on the structure and its function are warranted. A recent study shows

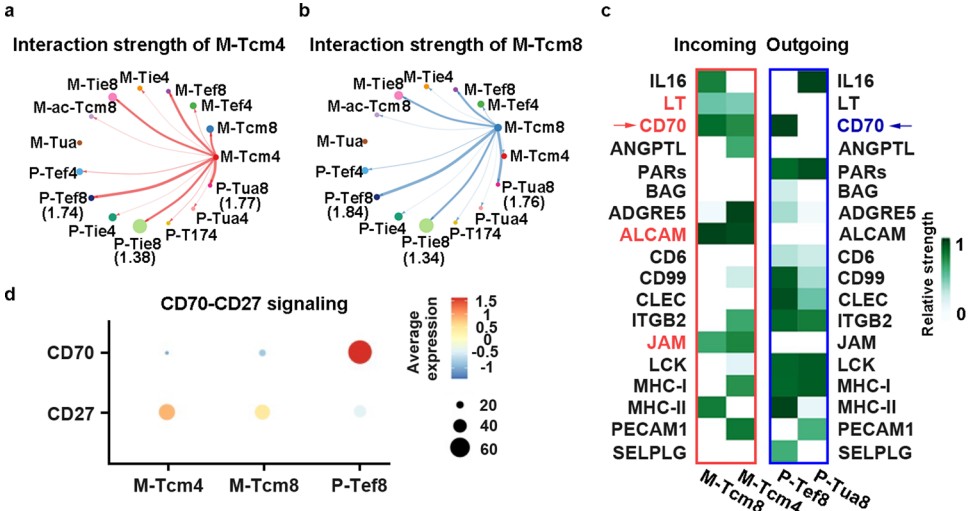

**Fig. 5 | Quantitative analysis of intercellular communication between CARMz T and CARP T cells. a**, **b** The inferred interaction strength of CD4+ central memory CARMz T cells (M-Tcm4) and CD8+ central memory CARMz T cells (M-Tcm8) with other cell clusters from CARMz T and CARP T cells. The top three interaction strength were shown with numbers that indicate the strength of interactions in Supplementary Table 1. **c** Heatmap shows the inferred incoming signaling pathways of M-Tcm4 and M-Tcm8 and outgoing signaling pathways of P-Tef8 and P-Tua8 by CellChat tool. Incoming signaling pathways are shown in the red box on the left and outgoing signaling pathways are shown in the blue box on the right. The relative strength was used to calculate intercellular communication probability. "M" indicates CARMz T and "P" indicates CARP T cells as for the name of each cell cluster on the horizontal axis. **d** Dot plot shows the average expression of CD27 and CD70 in M-Tcm4, M-Tcm8, and P-Tef8 clusters.

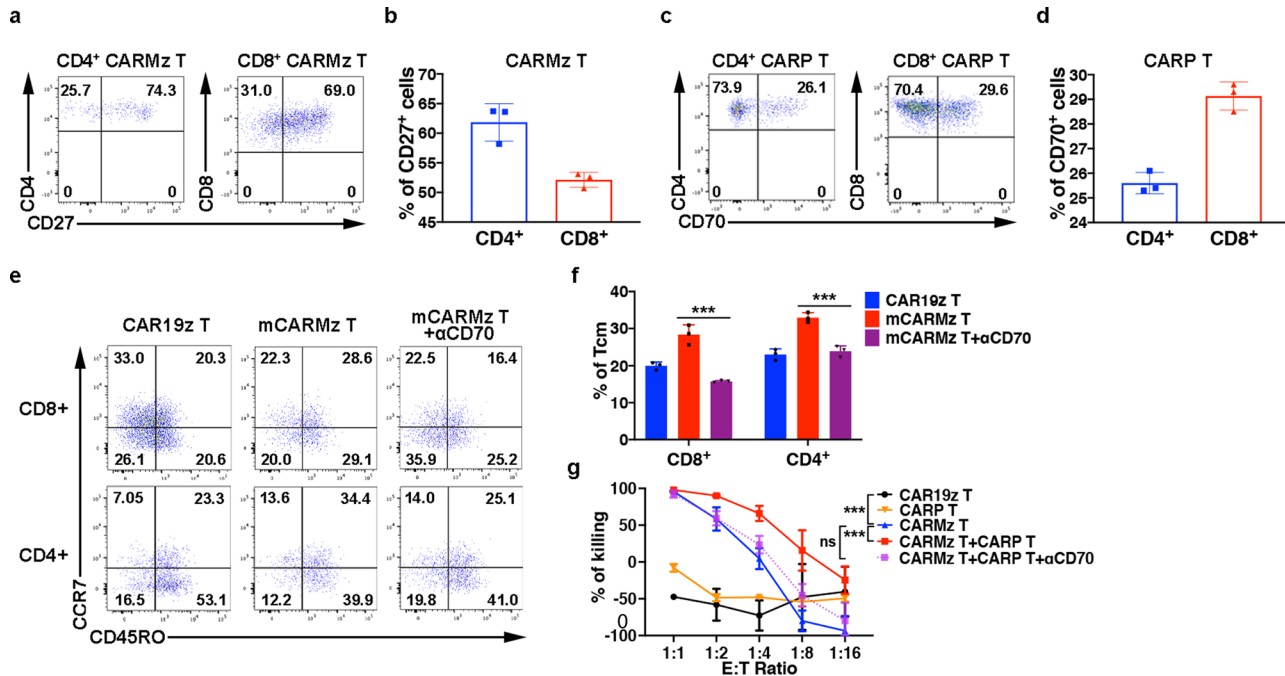

**Fig. 6 | CARP T cells augments the efficacy and persistence of CARMz T cells via the CD70-CD27 axis. a** Percentage of CD27 expression in CD8+ (gated on CD8+GFP+ cells) and CD4+ (gated on CD4+GFP+ cells) CARMz T cells post-coculture with HeLa-GL cells. **b** Bar graphs summarizing the percentage of CD27 expression in (**a**). Data are represented mean ± SD (N = 3 biological samples). **c** Percentage of CD70 expression in CD4+ (gated on CD4+CD19+ cells) and CD8+ (gated on CD8+CD19+ cells) CARP T cells post-coculture with HeLa-GL cells. **d** Bar graphs summarizing the percentage of CD70 expression in (**c**). Data are represented mean ± SD (N = 3 biological samples). **e** Representative phenotype of CARMz T cells (gated on CD8+GFP+ cells and CD8-GFP+ cells) from a mixture of CARMz T and CARP T (mCARMz T), a mixture of CARMz T and CARP T treated with anti-CD70 mAb (10 μg/ml) (mCARMz T + αCD70) and control CAR19z T cells were measured post-coculture with HeLa cells for 36 h. **f** Proportion of Tcm (central memory T cells, CD45RO+CCR7+) in (**e**). Data are presented as mean ± SD (N = 3 biological samples). p Values were calculated by two-way ANOVA with Tukey's multiple comparisons test (mCARMz T + αCD70 vs. mCARMz T = CD8+: 9.423E−07, CD4+: 2.858E−05). **g** CARP T, CARMz T, a mixture of CARMz T and CARP T, a mixture of CARMz T and CARP T treated with anti-CD70 mAb (αCD70) and control CAR19z T cells cytotoxicity against HeLa-GL cells were measured at various E:T ratios. Data are presented as mean ± SD (N = 3 independent experiments). p Values were calculated by two-way ANOVA with Tukey's multiple comparisons test (CARMz T vs. CAR19z T = 2.052E-09, CARMz T + CARP T vs. CARMz T = 7.427E−10, CARMz T + CARP T + αCD70 vs. CARMz T = 0.248). ***p < 0.001.

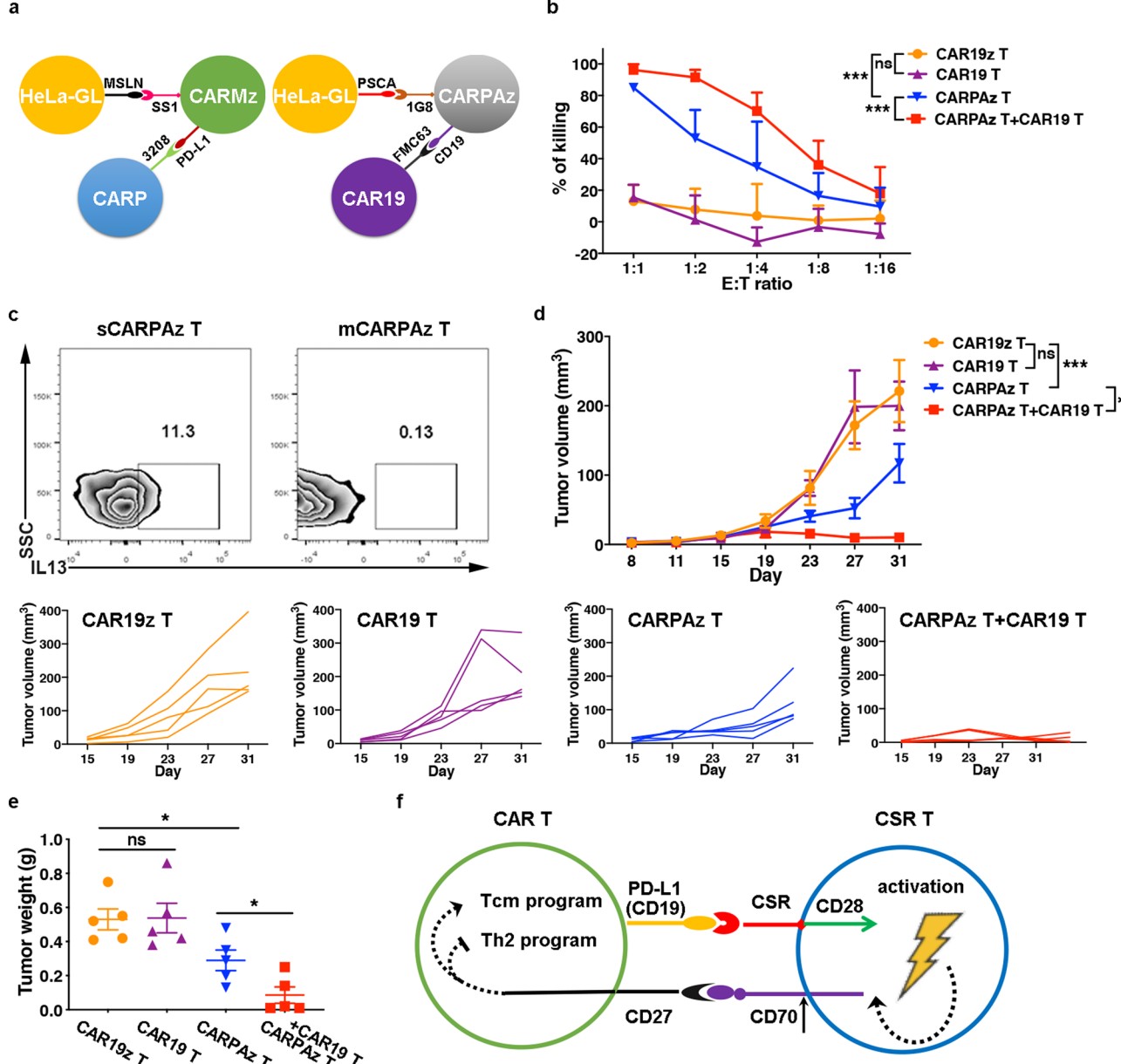

**Fig. 7 | CAR19 T cells enhance the antitumor effect of CARPAz T cells.**
**a** Schematic diagram of the heterotypic binding of CARMz T cells to CARP T cells
(left) and CARPAz T cells to CAR19 T cells (right). Antigens and scFvs are indicated.
**b** CAR19 T, CARPAz T, a mixture of CAR19 T and CARPAz T and control CAR19z
T cells cytotoxicity against targeting HeLa-GL cells were measured at various E:T
ratios. Data are presented as mean ± SD (*N* = 3 independent experiments). *p* Values
(CAR19 T vs. CAR19z T = 0.504, CARPAz T vs. CAR19z T = 1.591E-09, CARPAz T +
CAR19 T vs. CARPAz T = 2.471E-04). **c** Percentage of IL3⁺ cells in individual CARPAz
T (sCARPAz T) and CARPAz T from a mixture of CARPAz T and CAR19 T cells
(mCARPAz T) post-coculture with HeLa-GL cells at a 1:1 E:T ratio for 24 h (gated on
CD3⁺CD19⁺ cells). **d, e** NSI mice bearing HeLa-GL tumors (5 × 10⁵, established for
8 days) were infused with CARPAz T, CAR19 T, a mixture of CARPAz T and CAR19 T
or control CAR19z T cells (5 × 10⁶). **d** Tumor volumes were monitored on the

indicated days. Individual tumor responses to CAR-T cell injection are shown with
spider plots below. *p* Values (CAR19 T vs. CAR19z T = 0.999, CARPAz T vs. CAR19z
T = 5.491E-04, CARPAz T + CAR19 T vs. CARPAz T = 0.040). **e** Tumor weights were
measured after mouse euthanasia. *p* Values \ANOVA with Sidak's post hoc test
(CAR19 T vs. CAR19z T = 0.932, CARPAz T vs. CAR19z T = 0.019, CARPAz T + CAR19
T vs. CARPAz T = 0.042). **f** Schematic diagram of interactions between CSR T and
CAR-T cells. CSR T cells bound to CAR T cells through cell-cell contacts (CSR to PD-
L1 or CD19), promoted CAR T cells to differentiate to central memory-like T cells
(Tcm), and inhibited the formation of Th2 cells in CAR T cells via the CD70-CD27
axis. Data of **d** and **e** are presented as mean ± SEM (*N* = 5 mice per group). *p* Values
of **b** and **d** were calculated by two-way ANOVA with Tukey's multiple comparisons
test. **p* < 0.05 and ****p* < 0.001.

that homophilic interaction of CD56 on CAR T cells enhances anti-
tumor activity[63]. It is interesting to characterize whether a similar cell-
cell connection is formed through homophilic interaction of CD56
between two CAR T cells.

Previous studies show that the CD70-CD27 axis promotes T cell
survival and cytotoxicity, and the production and amplification of
virus-specific memory T cells[8,10]. Recent studies reports that anti-CD70
CAR T cells exhibit effective antitumor functions against renal

carcinoma and acute myeloid leukemia (AML)[64,65]. In addition, CD27
co-stimulation augments the survival and antitumor activity of CAR
T cells[66,67]. Based on scRNA-seq and FACS analysis, we found that CD70
was highly expressed in CD8⁺ effector CARP T cells and central
memory-like CARMz T cells expressed CD27 at high levels. In addition,
we found that the CD70-CD27 axis is essential for CARP T cells to
promote CARMz T cells to differentiate to central memory-like CARMz
T cells. Our results suggest that agonistic CD27 antibodies like

varlilumab may improve efficacy of CAR T cells[15,50]. Further studies are also warranted to identify any additional ligand-receptor interactions that contribute the effects of CARP T cells on CARMz T cells between these two types of cells.

In conclusion, our findings show that CARP T cells bound to CARMz T cells through cell-cell contacts, promoted CARMz T cells to differentiate to central memory-like T cells, and elevated their anti-tumor activity via the CD70–CD27 axis, indicating an important role of the trans-recognition between CSR and its antigen on CAR T cells in regulating the efficacy and persistence of CAR T cells (Fig. 7f).

## Methods

### Lentiviral vector design

The scFv (3208) for CARP was derived from atezolizumab (AZ), a high-affinity humanized antibody against PD-L1. The scFv for CAR19 was derived from FMC63, a commonly used scFv targeting CD19. Both CARP and CAR19 contained the CD28 costimulatory molecule (UniProt Entry P10747, aa 180-220) and truncated CD19 (tCD19) tag (GenBank NP_001171569.1, aa 11–323), without the CD3ζ chain in the cytoplasmic domain. The truncated CD19 only contained the extracellular domain and transmembrane domain of wild-type CD19. The construct CAR19z was consistent with CAR19 except that CAR19z contained the CD3ζ chain (UniProt Entry P20963, aa 52–164). CARMz contained a scFv (SS1) that targets mesothelin (MSLN), and CARPAz contained a scFv (1G8) that targets prostate stem cell antigens (PSCA), respectively. Both of them contained the CD28 costimulatory molecule and CD3ζ chains in the cytoplasmic domain. The GL vector is a reporter vector, which contains a firefly luciferase (luc) reporter gene (GenBank ABA41653.1, aa 1–550) and an enhanced GFP (eGFP) reporter gene (GenBank YP_009062989.1, aa 1–239). The MSLN-GL and PDL1-GL vector were obtained by adding the MSLN gene (GenBank NP_001170826.1, aa 1–622) and PD-L1 gene (GenBank NP_054862, aa 1–421) to the GL vector. DNA sequences were synthesized by GenScript Co., Ltd. (Nanjing, China) and cloned into the second-generation lentiviral vector pWPXLd.

### Lentivirus production

HEK-293T cells were co-transfected with the pWPXLd-based gene expression plasmid and two packaging plasmids (psPAX2 and pMD2. G) via PEI MAX 40 K (Polyscience, 24765-1). Lentivirus-containing supernatants were harvested at 48 and 72 h after transduction, filtered through a 0.45 μm filter, then immediately used or stored at 4 °C.

### CAR T cell manufacture

The healthy PBMC donors provided informed consent for the use of their samples for research purposes, and all procedures were approved by the Research Ethics Board of the Guangzhou Institutes of Biomedicine and Health. T cells were enriched from peripheral blood mononuclear cells (PBMCs) harvested from healthy donors using a pan T cell isolation kit (Miltenyi Biotec, 130-096-535) and then activated with a T cell activation and expansion kit (Miltenyi Biotec, 130-091-441) for 2 days in RPMI-1640 medium (Gibco, C11975500BT) supplemented with 10% fetal bovine serum (FBS) (Vigonob, XC6936T) and 1% penicillin/streptomycin (Gibco, 15140-122). CAR molecules were introduced to T cells through incubation with lentiviral supernatants in the presence of 8 μg/ml polybrene (Sigma-Aldrich, TR-1003-G) following T cell activation, and the medium was replaced with fresh medium containing IL-2 (300 IU/ml, PeproTech, AF-200-02) 12 h later. Subsequently, fresh medium was added every 1–2 days to maintain the cell density within the range of 0.5–1 × 10^6/ml.

### Cells and culture conditions

HEK-293T cells (ATCC: CRL-1573) were maintained in Dulbecco's modified Eagle's medium (DMEM) (Gibco, C11995500BT) supplemented with 10% FBS and 1% penicillin and streptomycin. HeLa (ATCC: CCL-2) is a MSLN+, PSCA+ and PD-L1− human cervical cancer cell line, H460 (ATCC: HTB-177) is a MSLN− and PD-L1+ human non-small-cell lung cancer cell line, NALM6 (ATCC: CRL-3273) is a CD19+ human acute lymphoblastic leukemia type B cell line and K562 cells (ATCC: CCL-243) a leukemia cell line that was negative for MHC-I molecules. All of them were obtained from ATCC and maintained in RPMI-1640 medium supplemented with 10% FBS and 1% penicillin and streptomycin. GL-expressing cell lines and H460-MSLN-GL cell line were generated through lentiviral transduction of the GL reporter gene or MSLN-GL gene into the parental cell lines, followed by sorting for GFP expression on a FACS AriaII cell sorter (BD Biosciences, San Jose, CA, USA). All cells were cultured at 37 °C in an atmosphere of 5% carbon dioxide.

### Flow cytometry

Flow cytometry was performed on a BD LSR Fortessa or Canto II, and the data were analyzed using FlowJo software (version 10.4.0). Cell-surface staining was performed by pelleting cells and resuspending them in 50 μl of FACS buffer (2% FBS in PBS) with antibodies for 30 min on ice in the dark. For intracellular IL13 staining, cells were fixed with Phosflow™ Fix Buffer I (BD, 557870) and permeabilized with Phosflow™ Perm Buffer III (BD, 558050). The protein transport inhibitor cocktail (Thermo Fisher, 00-4980) was used to prevent IL13 transport to extracellular space 6 h before flow cytometry analysis. Peripheral blood, spleen and tumor samples from xenograft mice were treated with red blood cell lysis buffer (BioLegend, 420301) before staining. Cells were washed with FACS buffer before analysis. The antibodies used in this research included in Supplementary Table 2.

### In vitro killing assays

Target tumor cells K562-PDL1-GL, HeLa-GL, and H460-MSLN-GL (10^4 cell/well) were incubated with CAR T cells or negative control T cells at the indicated E:T ratios in triplicate wells of U-bottomed 96-well plates at 37 °C for 24 h. Residual tumor cells were quantified by bioluminescent imaging (BLI) of the plate after adding 100 μl/well D-luciferin (potassium salt) (YeaseN, 40901ES03) at 150 μg/ml. The percentage of viable cells was equal to the experimental signal/maximal signal×100, and the percentage of cell lysis was equal to 100-percentage of viable cells. Both anti-PD-L1 mAb (atezolizumab) (Selleck, A2004) and anti-PD-1 mAb (Pembrolizumab) (Selleck, A2005) used at 20 μg/ml concentration. Anti-IL2 mAb (Biolegend, 500301), anti-IFN-γ mAb (ThermoFisher, 16-7318-81) and anti-CD70 mAb (Abcam, ab213102) used at 10 μg/ml concentration.

### Cytokine release assays

CAR T cells or control T cells were cocultured with target cells (K562-PDL1-GL, HeLa-GL or H460-MSLN-GL cells) at a 1:1 E:T ratio for 24 h, and then the culture supernatants were collected. The production of IL-2 (ThermoFisher, BMS221-2), IFN-γ (ThermoFisher, BMS228), IL5 (ThermoFisher, BMS278), IL10 (ThermoFisher, BMS215-2) and IL13 (ThermoFisher, BMS231-3) was measured with enzyme-linked immunosorbent assay (ELISA) kits. All ELISAs were performed according to the manufacturer's protocols.

### Confocal immunofluorescence

CARMz T and CARP T cells were separated based on their GFP and CD19 tag on a BD FACS Aria II platform. Purified CARMz T and CARP T cells were incubated with HeLa WT cells in a ratio of 4:1 for 36 h at 37 °C. Then CARMz T Cells (1 × 10^5) were harvested and added to Nunc Glass Bottom Dishes (Thermo-Scientific, 150680) with CARP T cells (1 × 10^5) and incubated for 30 min before applying live cell membrane stain at 37 °C for 30 min (CellBrite™ Steady 550 Membrane Staining Kits, 30107-T). CAR T cells were then stained in PD-L1 (Invitrogen, 4347834) and Hoechst (Yessen, 40731ES10) for 15 min at 37 °C. Fresh medium (DMEM + 10%FBS) was added

before imaging. CAR T cells were examined using an Andor Dragonfly 200 confocal microscope (Oxford instruments), in a $CO_2$ and temperature-controlled environmental chamber (Tokai Hit, Japan). Image analysis was conducted using Imaris 9.3 software (Bitplane, Oxford).

## Xenograft models and in vivo assessment

All animal experiments were performed based on an animal protocol approved by the relevant institutional animal care and use committee (IACUC) of Guangzhou Institutes of Biomedicine and Health. All mice used in these studies were aged 6–8 week-old NOD-SCID-IL2Rg$^{-/-}$ (NSI) mice[68]; both males and females were used. All experimental mice were co-housed within specific pathogen-free (SPF)-grade cages and provided autoclaved food and water, with a 12 h light/dark cycle and a temperature range of 21–27 °C with 40–60% humidity. HeLa-GL or H460-MSLN-GL cells in 200 μl of PBS were injected in the flank to establish subcutaneous tumors. Then, $2.5 \times 10^6$ or $5 \times 10^6$ CAR T cells or control T cells were adoptively transferred into the tumor-bearing mice via tail vein injection at the indicated time during each experiment. Tumors were measured on the indicated days with a caliper to determine the subcutaneous growth rate. Tumor volume was calculated using the following equation: (length × width$^2$)/2. The maximum size tumors allowed by the IACUC of Guangzhou Institutes of Biomedicine and Health is 2000 mm$^3$ and we have adhered to these size limits in all animal experiments. Mice were euthanized when tumor growth >2000 mm$^3$, using carbon dioxide asphyxiation followed by cervical dislocation.

We next selected a MSLN and PD-L1 double positive primary non-small cell lung cancer (NSCLC) sample that confirmed by immunohistochemistry (IHC) to develop first-generation NSCLC patient-derived xenograft (PDX) models. The primary NSCLC sample was from a 59-year-old male with poorly differentiated squamous cell carcinoma (stage IIIa) without EGFR mutation. This volunteer provided written informed consent, and the use of human material have been approved by the Research Ethics Board of Guangzhou Institutes of Biomedicine and Health. Briefly, surgical tumor samples were cut into $2 \times 2 \times 2$ mm$^3$ pieces and transplanted subcutaneously into the right flank of NSI mice. After the subcutaneous tumors reached an approximate size of 1000 mm$^3$, they were removed and transferred to secondary recipients to establish second-generation PDX models. Then, 10 days after tumor transplantation, the mice received with $5 \times 10^6$ CAR T cells or control T cells. Tumors were measured with a caliper, and tumor volume was calculated using the following equation: (length × width × width)/2.

## Bulk RNA-seq

Individual CARMz T (sCARMz T), individual CARP T (sCARP T), CARMz T (mCARMz T), and CARP T (mCARP T) from cocultures of CARMz T and CARP T and CARMz T cells treated with AZ (CARMz T + AZ) were separated by flow cytometry sorting based on their GFP or tCD19 tag post-coculture with HeLa-GL cells at 1:1 E:T ratio for 36 h, and then processed for bulk RNA-seq. Each sample had three biological replicates. Total RNA was extracted from the tissues using Trizol (Invitrogen, Carlsbad, CA, USA) according to the manufacturer's instructions, and sequencing was performed on a BGISEQ-500 (BGI, Wuhan, China). Sequence reads were trimmed for adaptor sequences, and low-complexity or low-quality sequences removed. The number of raw reads mapped to genes was calculated by RSEM (rsem-1.2.4), and the sample results were combined and normalized by EDAseq (1.99.1). Gene expression fold-changes were calculated using normalized raw reads. For downstream analysis, glbase scripts were used.

## scRNA sequencing cell capture and cDNA synthesis

Individual CARMz T (sCARMz T), CARMz T from a mixture of CARMz T and CARP T (mCARMz T), Individual CARP T (sCARP T) and CARP T

from a mixture of CARMz T and CARP T cells (mCARP T) were isolated by flow cytometry sorting (BD FACS AriaII) based on their GFP or tCD19 tag post-coculture with HeLa-GL cells at 1:1 E:T ratio for 36 h, and then processed for scRNA-seq. Using BD RhapsodyTM Cartridge Reagent Kit (BD, 633731) and BD RhapsodyTM Cartridge Kit (BD, 633733), the cell suspension (300–600 living cells per microliter) was loaded onto the RhapsodyTM Cartridge (BD) to generate single-cell magnetic beads in the microwells according to the manufacturer's protocol. In short, single cells were suspended in sample buffer (BD). Approximately 18,000 cells were added to each channel, allowing to recover an estimated number of 9000 target cells. Captured cells were lysed and the released RNA were barcoded through reverse transcription in individual microwells. Reverse transcription was performed on a ThermoMixer® C (Eppendorf) at 1200 rpm and 37 °C for 45 min. The cDNA was generated and then amplified, and quality assessed using an Agilent 4200. scRNA-seq libraries were constructed using BD RhapsodyTM WTA Amplification Kit (BD, 633801) according to the manufacturer's instruction, and finally sequenced using an Illumina Novaseq6000 sequencer with a sequencing depth of at least 50,000 reads per cell, with 150 bp paired-end (PE150) reads.

## Processing of scRNA-seq data

The BD Rhapsody analysis pipeline was used to process sequencing data (fastq files), the reference genome was GRCh38 (Ensembl). The scRNA-seq data was processed with the R package Seurat (version 3.1.5). Cells were removed if they had either fewer than 200 expressed genes or over 20% unique molecular identifiers (UMIs) originating from mitochondria. UMI counts were normalized and were log-transformed. Datasets were normalized and cell-cycle regression was performed. Two thousand highly variable Genes (HVGs) for each data were identified. Principal component analysis (PCA) and uniform manifold approximation and projection (UMAP) dimension reduction were performed with top 20 principal components. The Louvian modularity optimization algorithm was applied to iteratively group cells together into clusters, and cell clusters were visualized using UMAP and t-stochastic neighbor embedding (tSNE). Cell clusters were annotated to known biological cell types using canonical cell marker genes including T cell activation, proliferation, cytotoxicity, cytokines, chemokines and memory status. The intercellular communication among ligands, receptors, and their cofactors between CARMz T and CARP T cells were analyzed by CellChat tool[62].

## Statistical analysis

All statistical analyses were performed with Prism software (GraphPad, version 8.0). The statistical significance of differences between two groups of data was calculated by an unpaired t-test, ANOVA with Tukey's multiple comparison test and ANOVA with Sidak's post hoc test were used for multiple comparisons.

## Reporting summary

Further information on research design is available in the Nature Research Reporting Summary linked to this article.

# Data availability

All data are included in the Supplementary Information or available from the authors upon reasonable requests, as are unique reagents used in this Article. The Bulk RNA-seq data have been deposited in Sequence Read Archive (SRA) under accession codes SRR18804308, SRR18804309, SRR18804310, SRR18804311, SRR18804312, SRR18804313, SRR18804314, SRR18804315, SRR18804316, SRR18804317, SRR18804318, SRR18804319, SRR18804320, SRR18804321, SRR18804322. The single cell RNA-seq data have been deposited in SRA under accession codes SRR18750842 and SRR18750843. The FACS data have been deposited in the OMIX, China National Center for Bioinformation under accession code [OMIX001697].

The raw numbers for charts and graphs are available in the Source Data file whenever possible. Source data are provided with this paper.

## Code availability

No custom code was used in this study. The following publicly available pipelines were utilized for analysis: standard preprocessing and quality control of scRNAseq data was based on Seurat Guided tutorials [https://satijalab.org/seurat/articles/pbmc3k_tutorial.html], intercellular communication was based on CellChat toll [https://github.com/sqjin/CellChat].

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

## Acknowledgements

This study was supported by National Key Research and Development Plan, No. 2021YFE0202800 (P.L.), 2017YFE0131600 (Y.L.), 2019YFA0111500 (X.L.); National Natural Science Foundation of China, No. 82202031 (L.Q.), No. 81961128003 (P.L.), 81972672 (P.L.), 81870121 (P.L.), 81873847 (J.Y.) and 32170946 (Z.J.); The Youth Innovation Promotion Association of the Chinese Academy of Sciences (2020351, Z.J.); Guangdong Provincial Significant New Drugs Development, No. 2019B020202003 (P.L.); Guangdong Basic and Applied Basic Research Foundation, No. 2019A1515010062 (Y.Y.), 2020A1515011516 (X.W.), 2021A1515110005 (L.Q.), 2021A1515220077 (S.L.), 2020B1212060052; Science and Technology Program of Guangzhou, China (202002020083, X.L.); Guangzhou Medical University High-level University Construction Research Startup Fund, NO. B195002004013 (L.Q.); Open project of State Key Laboratory of Respiratory Disease, SKLRD-OP-202002 (Z.Z.); The University Grants Committee/Research Grants Council of the Hong Kong Special Administrative Region, China (Project No. AoE/M-401/20), acknowledge Innovation and Technology Fund (ITF); the Youth Talent Promotion project of Guangzhou Association for Science and Technology, No. X20210201015 (Yan-Lai Tang).

## Author contributions

P.Li and L.Q. conceived and designed the research study; L.Q., Y.C., T.Y., D.C., R.Z., S. L., Z.J., Q.W., Y. L., and S.W. performed the in vitro assays and animal experiments; L.Q., H.P., and Z.T. produced the CAR T cells; L.Q. and X.Li. performed and analyzed the bulk RNA-seq and scRNA-seq data; P.Li and L.Q. wrote the manuscript; J.P.T., P.Liu, W.W., J.Y., X.Luo., Z.Z., Y.Y., and D.Q. provided important research reagents and technical advice; J.P.T., R.W., and Q.T. provided critical advice on this study and revised the manuscript; and all authors revised and approved the manuscript.

## Competing interests

Z.T. and P.L. are scientific founders of GZI and have equity in GZI. The remaining authors declare no competing interests.

## Additional information

[1]China-New Zealand Joint Laboratory of Biomedicine and Health, State Key Laboratory of Respiratory Disease, Guangdong Provincial Key Laboratory of Stem Cell and Regenerative Medicine, Key Laboratory of Stem Cell and Regenerative Medicine, Guangzhou Institutes of Biomedicine and Health, Chinese Academy of Sciences, Guangzhou, China. [2]Key Laboratory of Biological Targeting Diagnosis, Therapy and Rehabilitation of Guangdong Higher Education Institutes, The Fifth Affiliated Hospital of Guangzhou Medical University, Guangzhou, China. [3]Guangzhou Laboratory, Guangzhou, China. [4]Centre for Regenerative Medicine and Health, Hong Kong Institute of Science & Innovation, Chinese Academy of Sciences, Hong Kong SAR, China. [5]Guangdong Zhaotai InVivo Biomedicine Co. Ltd, Guangzhou, China. [6]Guangdong Key Laboratory of Liver Disease Research, Key Laboratory of Liver Disease Biotherapy and Translational Medicine of Guangdong Higher Education Institutes, the Third Affiliated Hospital of Sun Yat-sen University, Guangzhou, China. [7]Guangdong Cord Blood Bank, Guangzhou, China. [8]Guangdong Women and Children Hospital, Panyu, Guangzhou, China. [9]Department of Paediatrics, the First Affiliated Hospital, Sun Yat-Sen University, Guangzhou, Guangdong, China. [10]Department of Radiology, Translational Provincial Education Department Key Laboratory of Nano-Immunoregulation Tumor Microenvironment, the Second Affiliated Hospital of Guangzhou Medical University, Guangzhou, China. [11]School of Biomedical Sciences, Stem Cell and Regenerative Medicine Consortium, Li Ka Shing Faculty of Medicine, The University of Hong Kong, Hong Kong SAR, China. [12]Cancer Immunotherapy Programme, Malaghan Institute of Medical Research, Wellington, New Zealand. [13]Bioland Laboratory (Guangzhou Regenerative Medicine and Health Guangdong Laboratory), Guangzhou, China. ✉e-mail: tjp@gzlab.ac.cn; li_peng@gibh.ac.cn

