## [Peer Review File · Nature Communications]

Reviewers' comments:

Reviewer #1 (Remarks to the Author):

In the article entitled "T cells expressing a chimeric switch receptor targeting PD-L1 augment the antitumor effects of CAR-T cells in solid tumors by promoting the differentiation of central memory-like T cells" the authors seek to (a) evaluate the effect of chimeric switch receptors (CSR) on targeted CAR-T cells using in vitro and in vivo experiments, and (b) characterize the changes in RNA expression induced by the addition of T cells bearing CSRs and CAR molecules, at both bulk and single-cell levels.

The authors report several findings. First, the addition of CARP T cells (expressing a CSR consisting of a PD-L1 receptor with CD28 transmembrane and intracellular domains) to CAR-T cells targeting Mesothelin (CARMz) enhances target cell killing, IL-2/IFN- γ release, and limits tumor size in in vivo models. Second, using both bulk and single-cell RNA-seq, they find that CARMz cells differentially express genes associated with central memory T cells after co-culture with the CARP T cells. Conversely, they next characterize transcriptional changes in CARP T cells and find differential expression of genes associated with effector T cells in CARP T cells co-cultured with CARMz T cells relative to CARP T cells cultured without. They conclude by evaluating the functionality of a separate CSR designed with an extracellular receptor specific for CD19 and again with the CD28 stimulatory transmembrane and intracellular domains. Using a CAR vector specific for PSCA (CARPaz T cells) they again find that combining the CSR-expressing T cells with CAR-T cells leads to smaller tumor volumes and weights in vivo, relative to infusion with either cell line alone.

Based on these results, they conclude that the combination of CSR expressing cells and CAR-T cells induces differentiation into central memory-like T cells. While this paper presents interesting preliminary findings, there are a substantial number of significant limitations and methodological concerns – primarily related to lack of validation and mechanisms and significant issues with the scRNA-seq analyses.

MAJOR COMMENTS:

1. The authors primary conclusion is that CSR expressing cells can promote a central memory-like T cell when paired with antigen-specific CAR-T cells. The data they present to support this conclusion is largely based on bulk RNA-seq and scRNA-seq differential gene expression analysis, both of which have several flaws. Regarding the scRNA-seq analysis:
 - a. The authors report using the "BD DataView software" in the Methods without providing further details. Typically, scRNA-seq data are filtered based on some per-cell quality metrics (number features per cell, mitochondrial content, doublets, complexity, etc.) and assessment of and correction of potential batch effects. No such pre-processing considerations are reported, raising significant concerns about the validity of the data.
 - b. It is unclear how the authors separate out the different cells from their scRNA-seq.
 - c. The authors determine CAR-T cell functional states based on cell clustering on a UMAP and a limited number of differentially expressed marker genes. Clustering is based on user-defined parameters and can be prone to overfitting, especially in samples that are biologically similar (e.g. all T-cells). While clustering can be useful for illustrating similarity or differences between cell types, it is not a substitute for more quantitative analyses (see below). In addition, clusters as defined by the authors are fairly indistinct with marker gene expression that is quite heterogeneous, more than I would expect for these methods.
 - d. The authors define subsets of T cells in both scRNA-seq analyses (Figures 3 and 4) using a limited set of marker genes, either plotting cluster enrichment on the UMAP image or in violin plots. This type of visual annotation is commonplace but not rigorous; it is more appropriate when multiple distinct cell types are present (e.g. in a tumor atlas). The authors should employ a method to score cells with a legitimate gene signature, rather than single genes, either from existing reference datasets or derived from their own data using dimension-reduction techniques (e.g. PCA or NMF).
 - e. The lack of NMF or a similar sort of clustering analysis represents a missed opportunity.
 - f. Methods for clustering distinct from the predetermined "BD DataView software" would be

important. tSNE would be important to show as an orthogonal approach.

2. The data presented in Figure 1 and Figure 5 are impressive, particularly the anti-tumor efficacy in the combined treatment group. However, other reports have already demonstrated that CSRs augment CAR-T cell efficacy in solid tumor models.^{1,2} Characterizing the T cell phenotypes in vivo at multiple time-points (or at least at the time of sacrificing the animal) represents an opportunity to validate the scRNA-seq findings – namely that CAR T cells adopt central memory phenotypes when combined with CARP T cells. This could also then facilitate pseudotime experiments which seem like an important analysis in this context. This could be performed via a number of potential experiments, including bulk RNA seq of the mouse tumor or flow-sorted mouse peripheral blood, IHC for relevant surface markers, or qPCR and Western blot for relevant T cell markers.

3. Can the authors obtain CAR T-cells from patients progressing on therapy and culture with the CARP T to show similar effect from a model closer to patients?

4. The Discussion section is brief, and introduces claims not substantiated by the data presented in the body of the Results.

a. In Lines 424- 431, the authors state, “Interestingly, we found that the binding of the FMC63 scFv on CAR19 T cells to CD19 on CARPAz T cells also enhanced the antitumor effect of a mixture of CARPAz and CAR19 T cells. Therefore, we speculated that the binding of PD-L1 and the 3208 scFv or the binding of CD19 and the FMC63 scFv can both decrease the distance between two types of cells and facilitate intercellular physical contact. This contact may be able to promote interactions between other cell-surface molecules or secreted paracrine factors and ultimately enhance the antitumor effects of CARMz or CARPAz T cells.” This line of reasoning is not explored in the body of the paper, yet comprises nearly half of the Discussion section.

b. Also, if the authors wish to pursue this line of reasoning, one logical next step would be to perform a ligand-receptor analysis between the CARMz cells and CARP cells, both of which have scRNA-seq data. While such an analysis would not be conclusive, it would provide a list of testable targets to further evaluate the mechanism of the cooperative signaling that appears to be occurring between CARP and CARMz cells.

5. The authors make lots of comments about secreted factors in response to cell stage changes but never show this with ELISA. Similarly, where is the data showing the co-cultured CARMz cells are functional – ELISpot seems like an important experiment here.

MINOR POINTS

1. The CAR19 abbreviation is introduced on Line 82 without prior definition. The same is true for CAR19z. The rationale for excluding/including the CD3ζ is not clear and should be explained.

2. The methods do not describe how the authors determined that PDX tumors were PD-L1+ MSLN+ positive.

3. The authors do not report any technical or biologic replicates of the bulk RNA-seq analysis, and they do not report their statistical approach for differential expression analysis.

4. In Lines 243-245, CAR19z T cells are mentioned. It becomes clear these cells are meant to be the negative control in some experiments, but this could be explicitly stated in this section to avoid confusion.

Reviewer #2 (Remarks to the Author):

This study is interesting and timely examining the mechanism underlying the ability of T cells expressing a chimeric switch receptor targeting PD-L1 to augment the efficacy of CAR-T cells. This paper explores the hypothesis that CSR efficacy comes from T cell interactions, targeting PD-L1 expression on the CAR-Ts. The authors present strong evidence that the combination of CSR T cells and CAR-T cells have at least additive if not synergistic antitumor effects. Further sequencing analyses demonstrate emergence of a central memory-like signature in CD8+ CAR-Ts and a Th1

signature in CD4+ CAR-Ts. Finally, the authors demonstrate that a different CSR/CAR-T combination had similar effects on IL-13 production and antitumor efficacy.

While the phenotypes are interesting, the manuscript in its current form is rather incomplete, and several conclusions are simply overstated and not supported by data. Addressing several key questions has the potential to enhance the quality of this study:

- 1) What mechanism/s allow CARP T cells alone to mediate tumor cell killing shown in Figs. 1B and 1D?
- 2) To what extent do the IL-2 and IFN γ produced by the CARP T cells affect the phenotype of the CARMz T cells? Would treatment of cocultures with tumor cells with anti-IL-2 or anti-IFN γ ablate the phenotypes observed?
- 3) The authors treat with AZ in Fig. 2A to test if soluble anti-PD-L1 would have the same effects as the CARP T cells, in the context of culture with a PD-L1- tumor cell line. Would anti-PD-1 have the same lack of effect?
- 4) Would pretreatment of CARMz T cells with AZ, which should block the interaction between CARP and CARMz T cells, ablate the effect of the co-culture?
- 5) Citations and suppositions for what phenotypes are for the cells from the sequencing analyses in figures 2, 3, and 4 should be supported by more formal GSEA analyses.
- 6) Importantly, what is the actual mechanism by which interaction between CSR T cells and CAR-T cells enhances CAR-T cell activity? This important point is not explored experimentally in the current manuscript.
- 7) Some of the conclusions are overstated and are not supported by the data presented. \For example, in discussing the results from Fig. 2, the authors state: "Taken together, these results suggested that CARP T cells prevented CARMz T cells from differentiating into immunosuppressive Th2 cells." This is based on sequencing data and on measurements of soluble IL-5, IL-10, and IL-13 from Fig. 2G. This statement seems to ignore the fact that CARMz T cells alone, as shown in Fig. 1, have robust antitumor efficacy.
- 8) Furthermore, the data presentation in Fig. 2G is misleading: the authors present Fold Change in these soluble factors, comparing with levels found in CARP T cultures with HeLa-GL cells, which the authors point out earlier do not express PD-L1. This is not a valid baseline. What are the actual pg/mL of these cytokines (as the authors display the data in panels 1C and 1E)? Are actual biologically significant amounts of these cytokines being produced?
- 9) Another glaring example of this is the discussion in Fig. 3D- the authors suggest based on the expression of several genes that CAR-T cells in coculture, after activation to introduce the CAR, are actually naive cells.

Reviewer #3 (Remarks to the Author):

The authors have performed a series of studies to test whether a CAR T mixture targeting various tumor targets and a CAR T cell that bridges the CAR T effector cell can enhance antitumor properties of the cell mixture compared to the primary CD28 based effector CAR T that is used in many laboratories. The results were obtained using a scFv on the CAR "helper" cell targeting PDL1 and also by expressing membrane bound CD19 in the CAR T "helper" with standard CD19 CAR T cells on the other side of the equation. The results are novel and could be translated into clinical trials.

The strengths are that the experiments are well performed technically, and the interpretations are appropriate for the data. Results are shown in vivo for solid tumors expressing msln and for

tumors expressing PSCA. The main weakness is that the mechanism is not completely defined at an immune synapse level. The efficacy could be further shown by infusing subtherapeutic doses of CAR T rather than testing at a single dose of 5×10^6 CA T cells per mouse.

Minor formatting issue is that the figure in Fig 2B, 3A, and 4A is not legible to me with the black on red text.

Response to Referees

Reviewers' comments:

Reviewer #1 (Remarks to the Author):

In the article entitled “T cells expressing a chimeric switch receptor targeting PD-L1 augment the antitumor effects of CAR-T cells in solid tumors by promoting the differentiation of central memory-like T cells” the authors seek to (a) evaluate the effect of chimeric switch receptors (CSR) on targeted CAR-T cells using in vitro and in vivo experiments, and (b) characterize the changes in RNA expression induced by the addition of T cells bearing CSRs and CAR molecules, at both bulk and single-cell levels.

The authors report several findings. First, the addition of CARP T cells (expressing a CSR consisting of a PD-L1 receptor with CD28 transmembrane and intracellular domains) to CAR-T cells targeting Mesothelin (CARMz) enhances target cell killing, IL-2/IFN- γ release, and limits tumor size in in vivo models. Second, using both bulk and single-cell RNA-seq, they find that CARMz cells differentially express genes associated with central memory T cells after co-culture with the CARP T cells. Conversely, they next characterize transcriptional changes in CARP T cells and find differential expression of genes associated with effector T cells in CARP T cells co-cultured with CARMz T cells relative to CARP T cells cultured without. They conclude by evaluating the functionality of a separate CSR designed with an extracellular receptor specific for CD19 and again with the CD28 stimulatory transmembrane and intracellular domains. Using a CAR vector specific for PSCA (CARPAz T cells) they again find that combining the CSR-expressing T cells with CAR-T cells leads to smaller tumor volumes and weights in vivo, relative to infusion with either cell line alone.

Based on these results, they conclude that the combination of CSR expressing cells and CAR-T cells induces differentiation into central memory-like T cells. While this paper presents interesting preliminary findings, there are a substantial number of significant limitations and methodological concerns – primarily related to lack of validation and mechanisms and significant issues with the scRNA-seq analyses.

MAJOR COMMENTS:

1. The authors primary conclusion is that CSR expressing cells can promote a central memory-like T cell when paired with antigen-specific CAR-T cells. The data they present to support this conclusion is largely based on bulk RNA-seq and scRNA-seq differential gene expression analysis, both of which have several flaws. Regarding the scRNA-seq analysis:

a. The authors report using the “BD DataView software” in the Methods without

providing further details. Typically, scRNA-seq data are filtered based on some per-cell quality metrics (number features per cell, mitochondrial content, doublets, complexity, etc.) and assessment of and correction of potential batch effects. No such pre-processing considerations are reported, raising significant concerns about the validity of the data.

Our response: We appreciate and agree the reviewer's comments. We have added the details of scRNA-seq procedure in the methods section (main text, page 19 line 23 to page 20 line 9) in the revised manuscript.

b. It is unclear how the authors separate out the different cells from their scRNA-seq.

Our response: Thanks for this comment. As we labeled CARMz T cells with GFP tag and CARP T cells with tCD19 tag separately (Fig. S1A), CARMz T and CARP T cells were separated by flow cytometry sorting based on their tag post co-cultured with HeLa-GL cells (Fig. 4A, S7A), then were subject to scRNA-seq experiments individually. We have provided this information in the methods section and figure legends of main text (page 19, lines 7-10; page 33, lines 1-5) and supplemental information (page 10 line 4 to page 11 line 4) of the revised manuscript.

c. The authors determine CAR-T cell functional states based on cell clustering on a UMAP and a limited number of differentially expressed marker genes. Clustering is based on user-defined parameters and can be prone to overfitting, especially in samples that are biologically similar (e.g. all T-cells). While clustering can be useful for illustrating similarity or differences between cell types, it is not a substitute for more quantitative analyses (see below). In addition, clusters as defined by the authors are fairly indistinct with marker gene expression that is quite heterogeneous, more than I would expect for these methods.

Our response: We apologize for the confusion. In fact, we used the R package Seurat (version 3.1.5) to analyze the scRNA-seq data. Clustering is unbiased and was analyzed based on PCA and UMAP dimension reduction methods. We have added this information in the main text (page 19 line 23 to page 20 line 9) of the revised manuscript. We will explain the clustering definition in the next question.

d. The authors define subsets of T cells in both scRNA-seq analyses (Figures 3 and 4) using a limited set of marker genes, either plotting cluster enrichment on the UMAP image or in violin plots. This type of visual annotation is commonplace but not rigorous; it is more appropriate when multiple distinct cell types are present (e.g. in a tumor atlas). The authors should employ a method to score cells with a legitimate gene signature, rather than single genes, either from existing reference datasets or derived from their own data using dimension-reduction techniques (e.g. PCA or NMF).

Our response: We appreciate the critics. According to the reviewers' suggestion, we have applied the Louvian modularity optimization algorithm to iteratively group cells together into clusters, and cell clusters were visualized using UMAP and tSNE (Figure S6A-B, main text, page 8, lines 11-15 and S7F-G, main text, page 10, lines 1-2). In addition, clustering is was analyzed based on PCA. Cell clusters were annotated to

known biological cell types using canonical cell marker genes related to T cell activation, cytotoxicity, proliferation, cytokines, chemokines, memory formation, based on following references (PMID: 31758530, 19538134, 31551493, 28218746, 29352091, 31017652, 16424171, 7989747, 32299851, 32655968, 26214741, 21926977). We have provided this information and cited corresponding references after each clustering definition (main text, page 20, lines 1-7) in the revised manuscript.

e. The lack of NMF or a similar sort of clustering analysis represents a missed opportunity.

Our response: We appreciate this critic. In fact, we have used PCA and UMAP dimension reduction for the clustering analysis of scRNA-seq data. This information has been supplemented in the revised manuscript (main text, page 20, lines 1-3).

f. Methods for clustering distinct from the predetermined “BD Data View software” would be important. tSNE would be important to show as an orthogonal approach.

Our response: Thanks for this advice. We have conducted scRNA-seq data analysis using two different approaches (UMAP and tSNE). We found that cell populations that were clustered using UMAP and tSNE dimension reduction methods were similar and consistent with each other. We have provided the results of analyses using tSNE in Figure S6A-B (main text, page 8, lines 11-13) and S7F-G (main text, page 10, lines 1-2) in the revised manuscript.

2. The data presented in Figure 1 and Figure 5 are impressive, particularly the anti-tumor efficacy in the combined treatment group. However, other reports have already demonstrated that CSRs augment CAR-T cell efficacy in solid tumor models.^{1,2} Characterizing the T cell phenotypes in vivo at multiple time-points (or at least at the time of sacrificing the animal) represents an opportunity to validate the scRNA-seq findings – namely that CAR T cells adopt central memory phenotypes when combined with CARP T cells. This could also then facilitate pseudotime experiments which seem like an important analysis in this context. This could be performed via a number of potential experiments, including bulk RNA seq of the mouse tumor or flow-sorted mouse peripheral blood, IHC for relevant surface markers, or qPCR and Western blot for relevant T cell markers.

Our response: We greatly appreciate the reviewer’s suggestions. Accordingly, we characterized the CARMz T cells by FACS when the mice were sacrificed and found that the percentages of central memory CAMz T cells infiltrated in tumor tissues of xenografts that were infused with the combination of CARMz T and CAPP T cells were higher than that from the CARMz T cell only group. We have shown the new results in Figure 1C-F (main text, page 5 line 24 to page 6 line 2) in the revised manuscript.

3. Can the authors obtain CAR T-cells from patients progressing on therapy and culture with the CARP T to show similar effect from a model closer to patients?

Our response: It is indeed a good idea. However, we cannot collect CAR-T cells from patients for this experiment due to the limitation of ethical issues. Instead, we

cocultured separated anti-CD19 CAR19z T cells (sCAR19z T) or a mixture of CAR19z T (mCAR19z T) and CARP T cells with CD19⁺ NALM6-GL cells (Fig. S8A, B) and found that the percentages of central memory T cells in CD8⁺ and CD4⁺ mCAR19z T cells were higher than those in CD8⁺ and CD4⁺ sCAR19z T cells, respectively (Fig. S8C, D). We have shown the results in Figure S8 (main text, page 12, lines 6-11) in the revised manuscript.

4. The Discussion section is brief, and introduces claims not substantiated by the data presented in the body of the Results.

a. In Lines 424- 431, the authors state, “Interestingly, we found that the binding of the FMC63 scFv on CAR19 T cells to CD19 on CARPAz T cells also enhanced the antitumor effect of a mixture of CARPAz and CAR19 T cells. Therefore, we speculated that the binding of PD-L1 and the 3208 scFv or the binding of CD19 and the FMC63 scFv can both decrease the distance between two types of cells and facilitate intercellular physical contact. This contact may be able to promote interactions between other cell-surface molecules or secreted paracrine factors and ultimately enhance the antitumor effects of CARMz or CARPAz T cells.” This line of reasoning is not explored in the body of the paper, yet comprises nearly half of the Discussion section.

Our response: We agree with the critics and have deleted these sentences in the discussion section of the revised manuscript. However, we indeed observed that CARMz T and CARP T cells formed immune synapse-like cell-cell contacts through PD-L1 on CARMz T cells and CSR on CARP T cells by confocal immunofluorescence. We have shown the results in Figure 2 (main text, page 6 line 23 to page 7 line 1) in the revised manuscript.

b. Also, if the authors wish to pursue this line of reasoning, one logical next step would be to perform a ligand-receptor analysis between the CARMz cells and CARP cells, both of which have scRNA-seq data. While such an analysis would not be conclusive, it would provide a list of testable targets to further evaluate the mechanism of the cooperative signaling that appears to be occurring between CARP and CARMz cells.

Our response: Many thanks for the excellent suggestion. Accordingly, we quantitatively analyzed intercellular communication networks based on scRNA-seq data of CARMz T and CARP T cells by CellChat tool (PMID: 33597522) and found that the CD8⁺ effector CARP T cells promoted CARMz T cells to differentiate to central memory-like T cells through the CD70-CD27 axis. In addition, the blockage of CD70-CD27 axis with anti-CD70 mAb (α CD70) reduced the percentages of central memory T cells and attenuated the tumor lysing capacity of CARMz T cells. We have shown the results in Figure 5 (main text, page 10 line 23 to page 12 line 4) in the revised manuscript.

5. The authors make lots of comments about secreted factors in response to cell stage changes but never show this with ELISA. Similarly, where is the data showing the co-cultured CARMz cells are functional – ELISpot seems like an important experiment here.

Our response: Thanks for the advice. Accordingly, we measured the concentrations of IL5, IL10, and IL13 in the supernatant of coculture of Hela-GL cells with CARMz T, CARP T, a mixture of CARMz T and CARP T with ELISA, and found that the levels of IL5, IL10 and IL13, three immune suppressive cytokines (PMID: 25687193, 25261204), were lower in the combination of CARMz T and CARP T cells, compared to that in CARMz T or CARP T cell cultures (Figure 3E) (main text, page 7, lines 24-27).

MINOR POINTS

1. The CAR19 abbreviation is introduced on Line 82 without prior definition. The same is true for CAR19z. The rationale for excluding/including the CD3 ζ is not clear and should be explained.

Our response: We apologize for the lack of definition. We have shown the structure of CAR19z and CAR19 in Figure S1A and explained their definition in details (main text, page 4, lines 25-26; main text, page 12, lines 17-20) in the revised manuscript.

2. The methods do not describe how the authors determined that PDX tumors were PD-L1+ MSLN+ positive.

Our response: Immunohistochemistry (IHC) staining was used to determine the expression levels of PD-L1 and MSLN in primary tumor samples. We have shown the results in Figure S5C (supplemental information, page 8, lines 9-10) and added experimental procedure information in the method section (main text, page 18, lines 12-14) in the revised manuscript.

3. The authors do not report any technical or biologic replicates of the bulk RNA-seq analysis, and they do not report their statistical approach for differential expression analysis.

Our response: We appreciate the critics. Accordingly, we have added biologic replicates of the bulk RNA-seq in Figure 3 and Figure S7 (main text, page 30, line 9; supplemental information, page 11, line 4) in the revised manuscript.

4. In Lines 243-245, CAR19z T cells are mentioned. It becomes clear these cells are meant to be the negative control in some experiments, but this could be explicitly stated in this section to avoid confusion.

Our response: We are sorry for the confusion. We have explicitly stated that CAR19z T cells served as negative control in the revised manuscript (main text, page 4, lines 25-26).

Reviewer #2 (Remarks to the Author):

This study is interesting and timely examining the mechanism underlying the ability of T cells expressing a chimeric switch receptor targeting PD-L1 to augment the efficacy of CAR-T cells. This paper explores the hypothesis that CSR efficacy comes from T cell interactions, targeting PD-L1 expression on the CAR-Ts. The authors present strong evidence that the combination of CSR T cells and CAR-T cells have at least

additive if not synergistic antitumor effects. Further sequencing analyses demonstrate emergence of a central memory-like signature in CD8+ CAR-Ts and a Th1 signature in CD4+ CAR-Ts. Finally, the authors demonstrate that a different CSR/CAR-T combination had similar effects on IL-13 production and antitumor efficacy.

While the phenotypes are interesting, the manuscript in its current form is rather incomplete, and several conclusions are simply overstated and not supported by data. Addressing several key questions has the potential to enhance the quality of this study: 1) What mechanism/s allow CARP T cells alone to mediate tumor cell killing shown in Figs. 1B and 1D?

Our response: Thanks for this comment. We think the mild tumor lysing by CARP T cells alone was not caused by CAR recognition, as HeLa-GL cells lack PD-L1 expression. The killing was possibly due to the HLA-mismatch between CARP T cells and HeLa-GL cells. Of note, CARP T cells suppressed tumor growth as poorly as CAR19z T cells, a negative control in xenografts (Fig. 1C, D). We repeated this *in vitro* killing with T cells from another donor and found the result showed that CARP T cells and control CAR19z T cells had comparable *in vitro* killing efficacy against HeLa-GL cells (Fig. 1A). The results from three biologic repeats were consistent. Regarding to the original Fig. 1D, H460-MSLN-GL cells express PD-L1, so they can inhibit the antitumor activity of T cells through PD-L1. As CARP molecule can rewire the inhibitory effects of PD-L1 into activating signals, CARP T cells showed better unspecific killing capacity against H460-MSLN-GL cells *in vitro* (Fig. S4B) and primary PD-L1+ NSCLC tumors *in vivo* (Fig. S5D, E) than CAR19z T cells.

2) To what extent do the IL-2 and IFN γ produced by the CARP T cells affect the phenotype of the CARMz T cells? Would treatment of cocultures with tumor cells with anti-IL-2 or anti-IFN γ ablate the phenotypes observed?

Our response: They are indeed very interesting questions. According to the suggestions, we treated co-cultures with tumor cells with anti-IL2 or anti-IFN γ antibodies and found that either anti-IL2 (10 ug/ml) or anti-IFN γ (10 ug/ml) did not attenuate the combination of CARP T and CARMz T cells to lyse tumor targets. The results were shown in Figure S4D (supplemental information, page 5 line 14 to page 6 line 4) in the revised manuscript.

3) The authors treat with AZ in Fig. 2A to test if soluble anti-PD-L1 would have the same effects as the CARP T cells, in the context of culture with a PD-L1- tumor cell line. Would anti-PD-1 have the same lack of effect?

Our response: We appreciate the suggestion. Accordingly, we treated the co-culture with anti-PD1 mAb (Pembrolizumab, 20 ug/ml) and found that blockage of PD-1 did not affect the *in vitro* killing efficiency of CARMz T cells. We have shown the results in Figure 2B (main text, page 29, lines 5-11) in the revised manuscript.

4) Would pretreatment of CARMz T cells with AZ, which should block the interaction between CARP and CARMz T cells, ablate the effect of the co-culture?

Our response: Yes, we found that pretreatment of CARMz T cells with AZ did ablate the effect of the co-culture (Figure 2C, main text, page 29, lines 11-18) in the revised manuscript.

5) Citations and suppositions for what phenotypes are for the cells from the sequencing analyses in figures 2, 3, and 4 should be supported by more formal GSEA analyses.

Our response: Thanks for this advice. We have performed GSEA analysis on both bulk RNA-seq and scRNA-seq data of sCARMz T and mCARz T cells, and shown these results in Figure 3D, 4F and 4G in the revised manuscript (main text, page 7, lines 22-27; main text, page 8 line 28 to page 9 line 2).

6) Importantly, what is the actual mechanism by which interaction between CSR T cells and CAR-T cells enhances CAR-T cell activity? This important point did not explored experimentally in the current manuscript.

Our response: We appreciate this critic, which has been raised by both Reviewer 1 and 2. We have further explored the mechanism by which interaction between CSR T cells and CAR-T cells enhances CAR-T cell activity and found that CARMz T and CARP T cells can form immune synapse-like cell-cell contacts through PD-L1 on CARMz T cells and CSR on CARP T cells by confocal immunofluorescence. In addition, we identified that the binding between CD70 on CARMz T cells and CD27 on CARP T cells promoted the formation of central memory-like T cells in CARMz T cells and augmented killing capacity of CARMz T cells. The results were shown in Figure 2 (main text, page 6 line 23 to page 7 line 1) and Figure 5 (main text, page 10 line 23 to page 12 line 4) of the revised manuscript.

7) Some of the conclusions are overstated and are not supported by the data presented. For example, in discussing the results from Fig. 2, the authors state: “Taken together, these results suggested that CARP T cells prevented CARMz T cells from differentiating into immunosuppressive Th2 cells.” This is based on sequencing data and on measurements of soluble IL-5, IL-10, and IL-13 from Fig. 2G. This statement seems to ignore the fact that CARMz T cells alone, as shown in Fig. 1, have robust antitumor efficacy.

Our response: We apologize for inaccurate statement. We have deleted it in the revised manuscript.

8) Furthermore, the data presentation in Fig. 2G is misleading: the authors present Fold Change in these soluble factors, comparing with levels found in CARP T cultures with HeLa-GL cells, which the authors point out earlier do not express PD-L1. This is not a valid baseline. What are the actual pg/mL of these cytokines (as the authors display the data in panels 1C and 1E)? Are actual biologically significant amounts of these cytokines being produced?

Our response: We agree with the critic. We have repeated the experiment and measured the concentrations of IL-5, IL-10 and IL-13 (pg/mL) in the supernatant of co-cultures by ELISA. The results were shown in Figure 3E (main text, page 7, lines 24-27) of the

revised manuscript.

9) Another glaring example of this is the discussion in Fig. 3D- the authors suggest based on the expression of several genes that CAR-T cells in coculture, after activation to introduce the CAR, are actually naive cells.

Our response: We are sorry for the inappropriate interpretation on Figure 3D. We have deleted the statement in the revised manuscript.

Reviewer #3 (Remarks to the Author):

The authors have performed a series of studies to test whether a CAR T mixture targeting various tumor targets and a CAR T cell that bridges the CAR T effector cell can enhance antitumor properties of the cell mixture compared to the primary CD28 based effector CAR T that is used in many laboratories. The results were obtained using a scFv on the CAR “helper” cell targeting PDL1 and also by expressing membrane bound CD19 in the CAR T “helper” with standard CD19 CAR T cells on the other side of the equation. The results are novel and could be translated into clinical trials.

The strengths are that the experiments are well performed technically, and the interpretations are appropriate for the data. Results are shown in vivo for solid tumors expressing msln and for tumors expressing PSCA.

The main weakness is that the mechanism is not completely defined at an immune synapse level.

Our response: We appreciate the critic. According to the advice, we used confocal immunofluorescence and observed that CARMz T and CARP T cells formed immune synapse-like cell-cell contacts through PD-L1 on CARMz T cells and CSR on CARP T cells in co-culture. In addition, we identified that the binding between CD70 on CARMz T cells and CD27 on CARP T cells promoted the formation of central memory-like T cells in CARMz T cells and augmented killing capacity of CARMz T cells. The results were shown in Figure 2 (main text, page 6 line 23 to page 7 line 1) and Figure 5 (main text, page 10 line 23 to page 12 line 4) of the revised manuscript.

The efficacy could be further shown by infusing subtherapeutic doses of CAR T rather than testing at a single dose of 5×10^6 CA T cells per mouse.

Our response: Thanks for this comment. Accordingly, we repeated the experiment by reducing the infusion dosages from 5×10^6 to 2.5×10^6 CAR-T cells per mouse. Similar results were obtained. In addition, we found that the percentages of central memory-like ($CD45RA^+CCR7^+$) CAMz T cells from the PBMC of xenografts infused with the combination of CARMz T and CAPP T cells were higher than those from the CARMz T cell group. We have shown the new results in Figure 1C-F (main text, page 5 line 24 to page 6 line 2) of the revised manuscript.

Minor formatting issue is that the figure in Fig 2B, 3A, and 4A is not legible to me with the black on red text.

Our response: We apologize for the mistakes. We have changed color scheme of Figure 3A, 4A and S7A in the revised manuscript.

REVIEWERS' COMMENTS

Reviewer #1 (Remarks to the Author):

Overall the authors have taken my comments seriously and addressed them in earnest. I have a few minor suggestions that appear to be overlooked, but otherwise would be supportive of publication.

1. The details now provides about scRNA-seq are helpful but the cutoff of 200 expressed genes is quite low. Are the data robust to a cutoff of 500 or 1000 genes?
2. As noted previously, the authors should really try NMF in addition to their PCA based approaches for demonstrating the robustness of their approach – these algorithms and code are available on github from the Tirosh group as well as other leading labs.
3. I appreciate the effort to show that the % central memory T-cells are increased in the combination condition. The authors did not perform pseudotime experiments I suggested, but I do not necessarily insist on this. I do feel it is a missed opportunity.

Reviewer #2 (Remarks to the Author):

The authors have addressed majority of my concerns/comments.
Just a few minor issues remain:

Include error bars in both directions on all their data. Figure 1c is missing error bars, and in Figure 2 it looks like in panels a-c there's only error bars on one side of the dots, and similar problems in Figures 6b and 6d, supplemental figure 3c and 3e, supplemental figure 4b, and supplemental figure 5a and 5d.

Reviewer #3 (Remarks to the Author):

Authors have addressed my comments and concerns. Interesting data on a potential mechanism is shown with the binding between CD70 on CARMz T cells and CD27 on CARP T cells.

Response to Referees

REVIEWERS' COMMENTS

Reviewer #1 (Remarks to the Author):

Overall the authors have taken my comments seriously and addressed them in earnest. I have a few minor suggestions that appear to be overlooked, but otherwise would be supportive of publication.

1. The details now provides about scRNA-seq are helpful but the cutoff of 200 expressed genes is quite low. Are the data robust to a cutoff of 500 or 1000 genes?

Our response: Many thanks for the suggestions. We have re-analyzed scRNA-seq data with the cutoff of 500 and 1,000 expressed genes, respectively. 4,264 individual CARMz T cells were collected with the cutoff of 200 expressed genes; 4,185 individual CARMz T cells were collected with the cutoff of 500 expressed genes; and 3,785 individual CARMz T cells were collected with the cutoff of 1,000 expressed genes. In addition, 3,924 individual CARP T cells were with the cutoff of 200 expressed genes; 3,829 individual CARP T cells were collected with the cutoff of 500 expressed genes; and 3,130 individual CARP T cells were collected with the cutoff of 1,000 expressed genes. These results suggest that the numbers of individual CARMz T or CARP T cells harvested are similar with different cutoff of expressed genes.

We next applied PCA and UMAP dimension reduction methods to process these scRNA-seq data. The Louvian modularity optimization algorithm was used to

iteratively group cells together into clusters. Cell clusters were then visualized using UMAP. These results below show that cell clusters and distribution of CARMz T (Fig. 1a-c) and CARP T cells (Fig. 1d-f) in UMAP were very similar with different cutoff of expressed genes, suggesting that the scRNA-seq data is also robust to a cutoff of 500 and 1000 expressed genes.

Figure 1. The UMAP plots of single CARMz T cells and CARP T cells.

(a-c) The UMAP plots of single CARMz T cells with the cutoff of 200 (a), 500 (b) and 1000(c) expressed genes. (d-e) The UMAP plots of single CARP T cells with the cutoff of 200 (a), 500 (b) and 1000 (c) expressed genes.

2. As noted previously, the authors should really try NMF in addition to their PCA based approaches for demonstrating the robustness of their approach, these algorithms and code are available on github from the Tirosh group as well as other leading labs.

Our response: Thanks for this advice. Eleven cell clusters were generated based on previous PCA analysis (Fig. 2a). Since cells in clusters 2 and 8 shared similar gene expression characteristics (high expressed CCR7 and LEF1) (Fig. 2b), we combined them together and named cluster Tcm8 in previous analysis (Fig. 2C). Similarly, cells in clusters 0, 3 and 10 were incorporated together and named Tie8 (Fig. 2a-c). We have performed NMF based approaches to analyze the scRNA-seq data of CARMz T cells accordingly. Eleven cell clusters were visualized in UMAP (Fig. 2d). Next we analyzed the correlation between the cell clusters generated by PCA analysis (Fig 2a) and the cell clusters produced by NMF analysis (Fig 2d). The coefficient heatmap shows that all cell clusters generated by PCA analysis are very similar to the ones generated by NMF analysis (Fig. 2e). In addition, corresponding cell clusters from PCA and NMF analysis shared very similar gene expression profiles (Fig. 2f). Next we defined each cell clusters according to their gene expression characteristics (Fig. 2g). Of note, non-activated T cells (Tua) and partially differentiated CD4⁺ effector T cells (Tie4) generated by PCA analysis (Fig. 2a-b) were incorporated together in the NMF analysis (Fig. 2e, g). Moreover, cell clusters 2 and 8 that we manually combined in PCA analysis and named cluster Tcm8 (Fig. 2a-b) were automatically classified into one cell cluster in NMF analysis (Fig 2e, g). Taken together, these results suggest that the results of PCA analysis were consistent with those of NMF analysis.

Figure 2. The NMF analysis of CARMz T cells

(a) The UMAP projection of CARMz T cells, cell clusters were generated by PCA analysis. (b) Dot plot of selected DEGs expressed in different clusters generated by PCA analysis. (c) The UMAP projection of CARMz T cells, cell clusters were generated by PCA analysis. (d) The UMAP projection of CARMz T cells, cell clusters were generated by NMF analysis. (e) Coefficients heatmap analysis for the correlation between the cell clusters generated by PCA analysis and the cell clusters produced by NMF analysis. (f) Dot plot of selected DEGs expressed in different clusters generated by NMF analysis. (g) The UMAP projection of CARMz T cells, cell clusters were generated by NMF analysis.

3. I appreciate the effort to show that the % central memory T-cells are increased in the combination condition. The authors did not perform pseudotime experiments I suggested, but I do not necessarily insist on this. I do feel it is a missed opportunity.

Our response: We are sorry that we did not perform pseudotime experiments before. We appreciate this suggestion and have performed monocle pseudotime analysis on CARMz T cells in clusters Tcm8 (CD8⁺ central memory T cells), Tef8 (CD8⁺ effector T cells), Tcm4 (CD4⁺ central memory T cells) and Tef4 (CD4⁺ effector T cells). The pseudotime analysis shows that cells in clusters Tef8 and Tef4 appeared at first. Then some of CD8⁺ and CD4⁺ effector T cells differentiated into cells in clusters Tcm8 and Tcm4, respectively (Fig. 3a-d). These analyses suggest that the cells in clusters Tcm8 and Tcm4 were central memory T cells, as they were derived from effector cells.

Figure 3. The pseudotime analysis of CARMz T cells in clusters Tcm8, Tef8, Tcm4 and Tef4.

(a-b) The pseudotime analysis of CARMz T cells in clusters Tcm8 and Tef8. (c-d) The pseudotime analysis of CARMz T cells in clusters Tcm4 and Tef4.

Reviewer #2 (Remarks to the Author):

The authors have addressed majority of my concerns/comments.

Just a few minor issues remain:

Include error bars in both directions on all their data. Figure 1c is missing error bars, and in Figure 2 it looks like in panels a-c there's only error bars on one side of the dots, and similar problems in Figures 6b and 6d, supplemental figure 3c and 3e, supplemental figure 4b, and supplemental figure 5a and 5d.

Our response: We apologize for the lack of error bars in figure 1C, 3E, 6D, supplemental figure 5A and supplemental figure 5D. We have added error bars of these figures in the

revised manuscript. In figure 2A-C, 6B, supplemental figure 3C and supplemental figure 4B, the data are represented as the mean \pm SEM. Therefore, the error bars are too small to be noticed. In the revised manuscript, we have changed mean with SEM to mean with SD in these figures.

Reviewer #3 (Remarks to the Author):

Authors have addressed my comments and concerns. Interesting data on a potential mechanism is shown with the binding between CD70 on CARMz T cells and CD27 on CARP T cells.

Our response: We greatly appreciate the reviewer's comments and suggestions.